# Towards Generalizable Detector for Generated Image

Qianshu Cai[1]    Chao Wu[2,3]    Yonggang Zhang[4]    Jun Yu[5]    Xinmei Tian[1] *

[1]MoE Key Laboratory of Brain-inspired Intelligent Perception and Cognition,
University of Science and Technology of China
[2]Zhejiang University    [3]School of Artificial Intelligence, Hebei Institute of Communications
[4]The Hong Kong University of Science and Technology
[5]The School of Intelligence Science and Engineering, Harbin Institute of Technology, Shenzhen

## Abstract

The effective detection of generated images is crucial to mitigate potential risks associated with their misuse. Despite significant progress, a fundamental challenge remains: ensuring the generalizability of detectors. To address this, we propose a novel perspective on understanding and improving generated image detection, inspired by the human cognitive process: Humans identify an image as unnatural based on specific patterns because these patterns lie outside the space spanned by those of natural images. This is intrinsically related to out-of-distribution (OOD) detection, which identifies samples whose semantic patterns (i.e., labels) lie outside the semantic pattern space of in-distribution (ID) samples. By treating patterns of generated images as OOD samples, we demonstrate that models trained merely over natural images bring guaranteed generalization ability under mild assumptions. This transforms the generalization challenge of generated image detection into the problem of fitting natural image patterns. Based on this insight, we propose a generalizable detection method through the lens of ID energy. Theoretical results capture the generalization risk of the proposed method. Experimental results across multiple benchmarks demonstrate the effectiveness of our approach. Code is available at `https://github.com/dav-joy-thon/DEnD-Detection`.

## 1   Introduction

In recent years, the development of generative AI has achieved significant breakthroughs. Specifically, diffusion-based generative technologies [22, 44, 10, 56] demonstrate revolutionary progress in the field of image synthesis. Advanced generative models, including Stable Diffusion [56], DALL-E 3 [50], Midjourney [41], and FLUX [29], enable users to generate highly realistic images through simple text prompts. Furthermore, the advanced video generation model, Sora [47], can produce high-definition, highly realistic videos and even simulate some physical effects. However, this rapid technological progression is not without its potential risks and challenges. The misuse of generated images for fraudulent purposes by malicious actors has sown seeds of doubt regarding the veracity of information in media sources. Therefore, it is crucial to develop an effective generated image detector with strong generalization capabilities to address these emerging threats.

In the realm of generated image detection, the majority of existing approaches are based on training binary classifiers [69, 68, 64]. For instance, DIRE [69] employs diffusion reconstruction error as an indicator to train a binary classifier. AEROBLADE [55] introduces a training-free approach that leverages the reconstruction error of an autoencoder to detect images generated by the Latent Diffusion Model (LDM) [56]. Unfortunately, AEROBLADE is limited to detecting LDM-generated images and necessitates access to the autoencoder used for image generation. These methods,

---

*Corresponding author (xinmei@ustc.edu.cn)

39th Conference on Neural Information Processing Systems (NeurIPS 2025).

however, confront a fundamental challenge: ensuring the generalizability of the constructed detector. In practical scenarios, generative models with unknown underlying architectures are frequently encountered, making the generalization challenge especially pronounced.

To address the generalization challenge, we revisit the process by which humans detect generated images. Humans who have only seen natural images can distinguish generated images with distinctive features. This could be attributed to the perception that the pattern of the generated image lies outside the space spanned by natural image patterns. In this regard, this out-of-space operation is also utilized in detecting out-of-distribution (OOD) data. Namely, OOD detectors should distinguish samples with semantic patterns, i.e., labels, that lie outside the semantic pattern space of in-distribution (ID) samples. The process by which humans recognize a generated image aligns with the principles of out-of-distribution (OOD) detection [73]. Humans have only seen natural images (ID) but can recognize generated images (OOD), and models have only seen ID samples but can detect OOD samples. This raises a fundamental yet under-explored question: can a model that has only seen natural images be used to detect generated images?

In this work, we propose a novel perspective: examining generated image detection through the lens of OOD detection. In this context, the pattern of natural images is regarded as ID data, while the pattern of generated images is OOD data. Building on the learnability theory of OOD detection [12], we study the generalizability of generated image detection, showing that models trained over natural images bring guaranteed generalization ability of generated image detection under mild assumptions. However, generated image detection relies on the disjoint space of specific patterns, while OOD detection focuses on the disjointed space of semantic labels for ID and OOD data. Namely, OOD detection can utilize classifiers trained on the label space of ID data, but generated image detection cannot use the classifiers trained for semantic labels.

To address this challenge, we draw inspiration from density-based and energy-based OOD detection. These methods highlight that the energy (density) of ID data is lower (higher) than that of OOD data. This is because models are trained to minimize (maximize) ID data's energy (density). Thus, we follow previous work and redefine the energy on ID data for generated image detection. Inspired by the seminal work in [61], we reveal that self-supervised models such as DINOv2 [48] exhibit latent capabilities [2] to discern pattern discrepancies between generated and natural images. Our theoretical results show that the learning objective of self-supervised models [34] is essentially to minimize the differential energy score of ID data, i.e., natural images. Based on this insight, we propose a novel framework, termed differential energy-based detection (DEnD), to discern generated images leveraging a pretrained self-supervised model, which demonstrates strong generalization capabilities.

Extensive experiments demonstrate that our method exhibits superior generalizability compared to training-based methods [69, 68, 64], and outperforms the state-of-the-art (SOTA) training-free methods. Our main contributions can be summarized as follows:

- We propose a novel view to understand and improve generated image detection by considering the natural image pattern as ID data and the generated image pattern as OOD data. In this context, we prove that models trained over natural images bring guaranteed generalization ability of generated image detection under mild assumptions.

- Drawing inspiration from energy-based OOD detection, we propose a novel framework termed differential energy-based detection (DEnD) to discern generated images, with theoretical guarantees on its generalizability.

- Comprehensive experimental results demonstrate that our DEnD framework not only surpasses the SOTA training-free method but also outperforms most training-based detectors. Moreover, our method exhibits remarkably strong generalization capabilities when faced with inaccessible generative models, e.g., Sora.

## 2  Related Work

**Advanced Generative Models.** Generative models have gained significant attention in recent years due to their ability to produce high-quality synthetic images. Generative Adversarial Networks (GANs) [16, 2, 26, 24] laid the groundwork for image generation. Following the advent of GANs,

---

[2]A detailed explanation of this insight is provided in Appendix A.

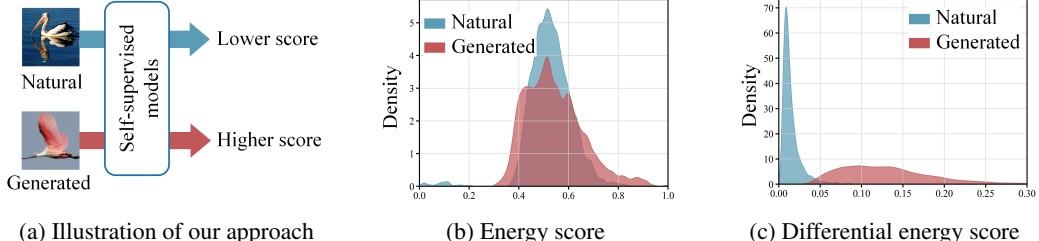

(a) Illustration of our approach   (b) Energy score   (c) Differential energy score

Figure 1: (a): We propose a score-based framework DEnD for generated image detection, where natural images exhibit lower differential energy scores and vice versa. (b): Directly detecting with energy-based OOD detection method yields suboptimal performance. (c): Inspired by the training objective of self-supervised learning, we porpose to detect generated images with differential energy score, which demonstrates strong generalization capabilities (see Appendix B for more details).

diffusion-based generative technologies [22, 44, 10] demonstrate revolutionary progress in the field of image synthesis. Recent advanced generative models such as Stable Diffusion [56], DALL-E 3 [50], Midjourney [41] and FLUX [29] demonstrate capabilities in creating detailed images from textual descriptions, marking a significant leap forward in generative capabilities.

**Generated Images Detection.** Early efforts on generated image detection primarily focused on color cues [39] and saturation cues [40]. However, with the emergence of ProGAN [15] and the diffusion model [22], these characteristics have become unreliable for detection purposes. Meanwhile, numerous frequency-based detection methods [28, 13, 58, 32] have also emerged. Mainstream training-based methods primarily focus on training a binary classifier network. CNNspot [68] for instance, manages to generalize a binary classifier trained on ProGAN to other architectures through the use of specific data augmentation techniques. DIRE leverages the reconstruction error from diffusion models to train classifiers. Nevertheless, training-based methods are often constrained by limited generalizability and the high costs associated with training. To mitigate these challenges, training-free methods have emerged. AEROBLADE computes the reconstruction error using an autoencoder in latent diffusion models to detect generated images, but its effectiveness is limited to LDMs. ZED [8] employs a lossless encoder pre-trained on natural images and leverages the coding cost gap to detect generated images. RIGID [20] exploits the differing robustness of real and AI-generated images to tiny noise perturbations within the representation space of vision foundation models. FSD [43] extracts forensic microstructures from images and models the distribution of real images using a Gaussian mixture model. On the foundation of these previous works, we revisit the generated image detection task through the lens of OOD detection and implement the DEnD framework, which demonstrates superior generalization capabilities with theoretical guarantees.

**Energy-based OOD Detection.** Out-of-Distribution (OOD) detection is a critical area of research that focuses on identifying data samples that differ significantly from the training distribution. Conventional approaches rely on confidence scores derived from softmax outputs [21]. However, neural networks can yield arbitrarily high softmax confidence for inputs that are substantially distant from the training data [42]. Energy-based OOD detection [33], on the other hand, maps each input to a scalar value that is lower for in-distribution (ID) data and higher for OOD data, thereby achieving superior performance. Theoretically, [12] has established the necessary conditions for the learnability of OOD detection and provided several sufficient conditions that characterize the learnability of OOD detection in specific practical scenarios. This theoretical foundation underpins our approach.

## 3 Preliminary

In this section, inspired by the human cognitive process, we formulate the generated image detection as an OOD detection task (see Sec. 3.1) and outline the primary objective of our paper (see Sec. 3.2).

### 3.1 Formulation

In this part, we elaborate on how to formulate generated image detection as an OOD detection task. Given a feature space of natural and generated images $\mathcal{X} \subset \mathbb{R}^d$ and two pattern spaces $\mathcal{T}_n := \{1\}$ to

represent the pattern of natural images, $\mathcal{T}_g := \{2\}$ to represent the pattern of generated images. We consider natural images as ID data and generated images as OOD data. Consequently, we have an ID joint distribution $D_{X_n T_n}$ over $\mathcal{X} \times \mathcal{T}_n$, where $X_n \in \mathcal{X}$ and $T_n \in \mathcal{T}_n$ are random variables. We also have an OOD joint distribution $D_{X_g T_g}$, where $X_g \in \mathcal{X}$ and $T_g \in \mathcal{T}_g$ are random variables. In empirical observations, natural and generated images are mixed in arbitrary and unknown proportions:

$$D_{XT} := (1 - \pi^{\mathrm{out}})D_{X_n T_n} + \pi^{\mathrm{out}}D_{X_g T_g}, \tag{1}$$

where the constant $\pi^{\mathrm{out}} \in [0, 1)$ is an unknown class-prior probability. We can only observe the marginal distributions:

$$D_X := (1 - \pi^{\mathrm{out}})D_{X_n} + \pi^{\mathrm{out}}D_{X_g}. \tag{2}$$

we define a subset of the function space as the hypothesis space $\mathcal{H} \subset \{h : \mathcal{X} \to \{1, 2\}\}$. 1 represents the natural images, and 2 represents the generated images. $h$ is called the hypothesis function. We explore the existence of a hypothesis space $\mathcal{H}$, such that for any joint distribution $D_{XT}$ belonging to the density-based space $\mathcal{D}_{XT}^{\mu,b}$ (see Appendix C.2), it satisfies generalizability (see Appendix C.1).

## 3.2 Objective

Our design objective can be described as follows: Using model $f$ trained over data $S$ to design a detector $g$, such that for any test data $\mathbf{x}$ drawn from the mixed marginal distribution $D_X$, the detector can differentiate whether the input is natural or generated. We define the differential energy score $\lambda$ (see Sec. 4.3). The detector classifies the data with lower scores as natural images and classifies the data with higher scores as generated images. The training data $S := \{\mathbf{x}^1, ..., \mathbf{x}^n\}$ is drawn independent and identically distributed from the joint distribution of natural images $D_{X_n}$.

# 4 Method

In this section, we first prove that the detector we have modeled in Sec. 3.1 exhibits generalizability under mild assumptions (see Sec. 4.1). Based on the theory, we then consider an advanced approach in OOD detection: energy-based OOD detection (see Sec. 4.2). However, experiments reveal that directly applying energy-based OOD detection yields suboptimal performance. Motivated by this observation and inspired by the training objectives of self-supervised learning , we present a generalizable training-free generated image detection framework, DEnD (see Sec. 4.3). We further provide theoretical guarantees (see Sec. 4.4) for the generalizability of our proposed method, establishing both its practical effectiveness and theoretical soundness.

## 4.1 Generalizability of Generated Detector

Despite the completion of our formulation, we cannot ascertain under what circumstances the resulting detector can generalize. We consider a significant assumption in learning theory—the Realizability Assumption (see Appendix C.3). This assumption implies that there exists at least one model in the hypothesis space $\mathcal{H}$ that can perfectly fit the training data, i.e., there are no classification errors. Under this assumption, we have a significant lemma derived from the learnability of OOD detection [12]:

**Lemma 4.1** *Given a density-based space $\mathcal{D}_{XT}^{\mu,b}$, if $\mu(\mathcal{X}) < +\infty$, the Realizability Assumption holds, then when $\mathcal{H}$ has finite Natarajan dimension [59], OOD detection is learnable in $\mathcal{D}_{XT}^{\mu,b}$ for $\mathcal{H}$.*

In our formulation, the generalizability of the detector and the learnability of OOD detection are equivalent. Therefore, we have derived several conditions for the detector's generalizability:

- $\mu(\mathcal{X}) < +\infty$, i.e., the feature space has a finite measure.
- An appropriate hypothesis space $\mathcal{H}$ is selected that satisfies the Realizability Assumption.
- The hypothesis space $\mathcal{H}$ we selected has a finite Natarajan dimension. This indicates that the model's complexity is controlled and that it is capable of generalizing well to unseen data.

## 4.2 Energy-based Detection

Since we have theoretically validated that the detector formulated as an OOD detection task is generalizable under mild assumptions, we consider whether we can simply and directly apply OOD

detection methods to effectively detect generated images. We consider an advanced method in OOD detection: energy-based OOD detection [33] [3].

We consider a discriminative neural classifier $q(\mathbf{x}) : \mathbb{R}^D \rightarrow \mathbb{R}^K$, which maps input $\mathbf{x} \in \mathbb{R}^D$ to logits. The energy-based OOD detection defines the free energy function $E(\mathbf{x}; q)$ over $\mathbf{x} \in \mathbb{R}^D$ as:

$$E(\mathbf{x}; q) = -\tau \cdot \log \sum_i^K e^{q_i(\mathbf{x})/\tau}. \tag{3}$$

$q_i(\mathbf{x})$ indicates the i-th index of $q(\mathbf{x})$. The temperature coefficient $\tau$ is treated as a hyperparameter.

Since the logit corresponding to the i-th label $q_i(\mathbf{x})$ can be expressed as $q_i(\mathbf{x}) = (f(\mathbf{x}), \mathbf{w}_i)$, we can rewrite the free energy function as:

$$E(\mathbf{x}) = -\tau \cdot \log \sum_i^K e^{(f(\mathbf{x}), \mathbf{w}_i)/\tau}, \tag{4}$$

where $f(\mathbf{x}) : \mathbb{R}^D \rightarrow \mathbb{R}^d$ indicates the feature extracted from the input $\mathbf{x}$ and $\mathbf{w}_i \in \mathbb{R}^d$ indicates the weight corresponding to the i-th label. We use $(\mathbf{a}, \mathbf{b})$ to denote the inner product of vectors $\mathbf{a}$ and $\mathbf{b}$. [33] theoretically prove that a model trained with negative log-likelihood (NLL) loss will push down energy for in-distribution data points. In the actual detection process, inputs with higher energies are naturally considered as OOD inputs and vice versa.

Unfortunately, our experiments reveal that both natural and generated images exhibit indistinguishable energy distributions (see Figure 1b), making it difficult for the detector to differentiate. This limitation arises because energy-based ood detection is typically derived from semantic label classifiers trained with NLL loss, whereas the discrepancy between natural and generated images manifests through divergences in high-level patterns rather than simplistic semantic label mismatches. To address this challenge, we propose replacing semantic label classifiers with models that capture non-label global patterns and redefine an energy function aligned with the model's training objective.

### 4.3 Differential Energy-based Detection

As demonstrated in [61], self-supervised models exhibit superior sensitivity to global characteristics compared to supervised models operating in label spaces. This property endows self-supervised models with latent capacities to discern pattern discrepancies between generated and natural images.

In self-supervised learning [18], a common approach is as follows: given a feature extractor $f(*)$ within a batch of size $N$, for an anchor $\mathbf{x}$, the positive sample is denoted as: $\mathbf{x}^+ = m(\mathbf{x})$, where $m(\mathbf{x})$ indicates the random transformation such as Gaussian blur. The other $N - 1$ samples are considered negative samples. Given the training sample $\mathbf{x}$, the loss function can be expressed as:

$$-\log \frac{e^{(f_\theta(\mathbf{x}), f_\theta(\mathbf{x}^+))/\tau}}{\sum_{i=0}^{N} e^{(f_\theta(\mathbf{x}), f_\theta(\mathbf{x}_i))/\tau}}, \tag{5}$$

where the notation $\mathbf{x}_0 = \mathbf{x}^+$ denotes the positive sample resulting from the random transformation.

In the context of self-supervised learning, each negative sample $\mathbf{x}_i$ is regarded as a distinct class within the discriminative model. Consequently, the features of the negative samples $f(\mathbf{x}_i)$ are akin to the weights $\mathbf{w}_i$ corresponding to their respective classes. By combining Equation 4 and Equation 5, we redefine our energy function [4] as follows:

$$E(\mathbf{x}; f) = \sum_{i=0}^{N} e^{(f(\mathbf{x}), f(\mathbf{x_i}))/\tau}. \tag{6}$$

The sum is for one positive sample and $N$ negative samples. In the self-supervised model $f(*)$ trained on ID data, the training objective can be expressed as:

$$\min_\theta \mathbb{E}_{\mathbf{x} \sim P_{\text{ID}}, m \sim \mathcal{M}} E(\mathbf{x}; f_\theta). \tag{7}$$

---

[3]Discussions on additional OOD detection approaches are provided in Appendix E.

[4]Energy function discussed hereafter adhere to the specific formulation presented in this section.

In our framework, the random transformation function $m(*)$ is treated as a random variable, which is drawn from a defined probability distribution $\mathcal{M}$. This distribution encapsulates the likelihood of various transformations being applied to the data.

We sample $k$ points from the random transformation distribution $\mathcal{M}$. Each sampled point represents a transformation function $m_i(*)$. The training objective can be rewritten as:

$$\min_{\theta} \mathbb{E}_{\mathbf{x} \sim P_{\text{ID}}} \left[ \frac{1}{k} \sum_{i}^{k} E_{m_i}(\mathbf{x}; f_\theta) \right].$$ (8)

The notation $E_{m_i}$ denotes the energy resulting from the random transformation $m_i(*)$.

Therefore, for the energy of $\mathbf{x}$ within the ID distribution, we can deduce [5] that for any $\epsilon > 0$:

$$\frac{1}{k} \sum_{i}^{k} |E(\mathbf{x}; f) - E(m_i(\mathbf{x}); f)| < \epsilon.$$ (9)

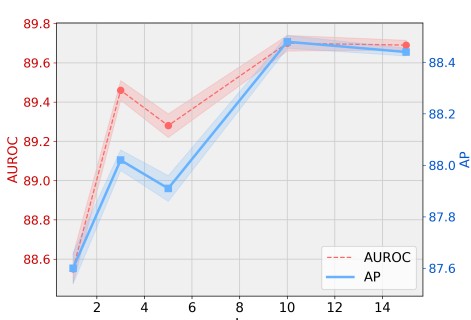

Figure 2: Impact of sampling frequency from $\mathcal{M}$ on DEnD's performance.

As depicted in Figure 2, in our experiments, we tested on ImageNet [9] with sampling from the random transformation distribution $\mathcal{M}$ at 1, 3, 5, 10, and 15 times. To strike a balance between accuracy and computational cost, we opted for a single sampling.

Hence, we can deduce that for any $\epsilon > 0$ any ID data $\mathbf{x}$, i.e., natural images, the following holds:

$$|E(\mathbf{x}; f) - E(m(\mathbf{x}); f)| < \epsilon.$$ (10)

Based on this approach, we obtained the differential energy score as follows:

$$\lambda(\mathbf{x}; f, m) = |E(\mathbf{x}; f) - E(m(\mathbf{x}); f)|.$$ (11)

As derived in Equation 10, the training objective of self-supervised models can be formulated as minimizing the differential energy scores for ID data (natural images). Consequently, for $\mathbf{x}$ drawn from the in-distribution (ID), which corresponds to natural images, $\lambda(\mathbf{x}; f, m)$ is relatively small, as shown in Figure 1c. Given the discrimination ability of the differential energy score, we employ it in generated image detection:

$$g(\mathbf{x}; \gamma, m, f) = \begin{cases} 1(natural) & \text{if } \lambda \leq \gamma, \\ 2(generated) & \text{if } \lambda > \gamma, \end{cases}$$ (12)

where $\gamma$ is the threshold [6] and $f$ denotes the pre-trained self-supervised model. In practice, we employ the powerful self-supervised Vision Transformer (ViT) model DINOv2 (see Appendix I.2 for detailed selection of self-supervised models), which is pretrained on an exceedingly vast dataset of natural images. Images that exhibit higher differential energy scores are classified as generated, while those with lower differential energy scores are classified as natural.

## 4.4 Generalizability of DEnD

In this section, building upon Sec. 4.1, we theoretically ground the Differential Energy-based Detection (DEnD) framework, demonstrating its generalizability for generated image detection.

In Sec. 4.1, we state that to ensure the detector's generalizability, the hypothesis space must satisfy the Realizability Assumption. Therefore, we first validate that our proposed DEnD adheres to this assumption. Our method designs $\mathcal{H}^*$ comprising a score-based classifier (see Figure 1a):

$$h_\gamma(\mathbf{x}) = \begin{cases} 1 & \text{if } \lambda(\mathbf{x}; f, m) \leq \gamma. \\ 2 & \text{if } \lambda(\mathbf{x}; f, m) > \gamma. \end{cases}$$ (13)

---

[5]See Appendix D for complete derivation.

[6]For more detailed explanations regarding the threshold, please refer to Appendix F.

Table 1: The performance of various detectors on ImageNet. The **bolded** text represents the best performance, and the underlined text represents the second-best performance.

| Methods | ADM | | ADMG | | LDM | | DiT | | Models BigGAN | | GigaGAN | | StyleGAN XL | | RQ-Transformer | | Mask GIT | | Average | |
|---|---|---|---|---|---|---|---|---|---|---|---|---|---|---|---|---|---|---|---|---|
| | AUROC | AP | AUROC | AP | AUROC | AP | AUROC | AP | AUROC | AP | AUROC | AP | AUROC | AP | AUROC | AP | AUROC | AP | AUROC (↑) | AP (↑) |
| | | | | | | | | | Training-based Methods | | | | | | | | | | | |
| CNNspot | 62.25 | 63.13 | 63.28 | 62.27 | 63.16 | 64.81 | 62.85 | 61.16 | 85.71 | 84.93 | 74.85 | 71.45 | 68.41 | 68.67 | 61.83 | 62.91 | 60.98 | 61.69 | 67.04 | 66.78 |
| UnivFD | 83.37 | 82.95 | 79.60 | 78.15 | 80.35 | 79.71 | 82.93 | 81.72 | 93.07 | 92.77 | 87.45 | 84.88 | 85.36 | 83.15 | 85.19 | 84.22 | 90.82 | 90.71 | 85.35 | 84.25 |
| DIRE | 51.82 | 50.29 | 53.14 | 52.96 | 52.83 | 51.84 | 54.67 | 55.10 | 51.62 | 50.83 | 50.70 | 50.27 | 50.95 | 51.36 | 55.95 | 54.83 | 52.58 | 52.10 | 52.70 | 52.18 |
| NPR | 85.68 | 80.86 | 84.34 | 79.79 | **91.98** | 86.96 | 86.15 | 81.26 | 89.73 | 84.46 | 82.21 | 78.20 | 84.13 | 78.73 | 80.21 | 73.21 | 89.61 | 84.15 | 86.00 | 80.84 |
| DRCT | 90.26 | 90.07 | 85.74 | 83.85 | 90.24 | **89.88** | **88.27** | **89.06** | 95.87 | 94.99 | 86.89 | 86.12 | 89.11 | 88.39 | 92.38 | 92.41 | 94.44 | 94.47 | 90.36 | 89.92 |
| | | | | | | | | | Training-free Methods | | | | | | | | | | | |
| AEROBLADE | 55.61 | 54.26 | 61.57 | 56.58 | 62.67 | 60.93 | 85.88 | 87.71 | 44.36 | 45.66 | 47.39 | 48.14 | 47.28 | 48.54 | 67.05 | 67.69 | 48.05 | 48.75 | 57.87 | 57.85 |
| RIGID | 87.00 | 85.29 | 81.22 | 77.90 | 74.60 | 69.51 | 70.22 | 67.17 | 87.81 | 86.23 | 85.54 | 84.39 | 86.58 | 86.41 | 90.66 | 89.89 | 89.94 | 88.41 | 83.73 | 81.69 |
| DEnD (ours) | **96.94** | **95.74** | **90.15** | **86.80** | 91.03 | 88.52 | 81.74 | 77.86 | **99.85** | **99.87** | **98.10** | **97.53** | **97.47** | **96.24** | **99.19** | **98.78** | **98.84** | **98.63** | **94.81** | **93.33** |

Table 2: The performance of various detectors on LSUN-BEDROOM. The **bolded** text represents the best performance, and the underlined text represents the second-best performance.

| Methods | ADM | | DDPM | | iDDPM | | Models Diffusion GAN | | Projected GAN | | StyleGAN | | Unleashing Transformer | | Average | |
|---|---|---|---|---|---|---|---|---|---|---|---|---|---|---|---|---|
| | AUROC | AP | AUROC | AP | AUROC | AP | AUROC | AP | AUROC | AP | AUROC | AP | AUROC | AP | AUROC (↑) | AP (↑) |
| | | | | | | | Training-based Methods | | | | | | | | | |
| CNNspot | 64.83 | 64.24 | 79.04 | 80.58 | 76.95 | 76.28 | 88.45 | 87.19 | 90.80 | 89.94 | 95.17 | 94.94 | 93.42 | 93.11 | 84.09 | 83.75 |
| UnivFD | 71.26 | 70.95 | 79.26 | 78.27 | 74.80 | 73.46 | 84.56 | 82.91 | 82.00 | 78.42 | 81.22 | 78.08 | 83.58 | 83.48 | 79.53 | 77.94 |
| DIRE | 57.19 | 56.85 | 61.91 | 61.35 | 59.82 | 58.29 | 53.18 | 53.48 | 55.35 | 54.93 | 57.66 | 56.90 | 67.92 | 68.33 | 59.00 | 58.59 |
| NPR | 75.43 | 72.60 | 91.42 | 90.89 | 89.49 | 88.25 | 76.17 | 74.19 | 75.07 | 74.59 | 68.82 | 63.53 | 84.39 | 83.67 | 80.11 | 78.25 |
| DRCT | 74.59 | 71.37 | 85.45 | 84.98 | 87.17 | 86.99 | 94.19 | 94.16 | 95.96 | 95.67 | 93.92 | 94.66 | 89.51 | 89.07 | 88.68 | 88.13 |
| | | | | | | | Training-free Methods | | | | | | | | | |
| AEROBLADE | 57.05 | 58.37 | 61.57 | 61.49 | 59.82 | 61.06 | 47.12 | 48.25 | 45.98 | 46.15 | 45.63 | 47.06 | 59.71 | 57.34 | 53.85 | 54.25 |
| RIGID | 69.76 | 68.31 | 88.35 | 88.82 | 84.15 | 84.54 | 91.85 | 92.28 | 92.65 | 93.18 | 78.09 | 76.54 | 91.94 | 92.28 | 85.25 | 85.13 |
| DEnD (ours) | **85.14** | **82.24** | **97.16** | **96.07** | **95.46** | **93.96** | **99.22** | **99.06** | **99.45** | **99.32** | **96.75** | **95.72** | **99.17** | **98.84** | **96.05** | **95.03** |

The design of DEnD exploits the property that $f$ results in relatively low $\lambda(\mathbf{x}; f, m)$ for natural images and high values for generated images, driven by the training objective of the self-supervised models. This separation underpins the following theorem:

**Theorem 4.2** *If there exists a threshold $\gamma' \in \mathbb{R}$ satisfying:*

$$\sup_{\mathbf{x} \in suppD_{X_n}} \lambda(\mathbf{x}; f, m) < \gamma' < \inf_{\mathbf{x} \in suppD_{X_g}} \lambda(\mathbf{x}; f, m), \tag{14}$$

*the hypothesis space $\mathcal{H}^*$ fulfills the Realizability Assumption, where supp means the support set.*

The proof is detailed in Appendix C.4. The theorem demonstrates that, owing to the discriminative power of our method, the differential energy scores between natural and generated images are separable and the Realizability Assumption holds. This provides critical assurance for the generalizability of our method. Building on the adherence of DEnD to the Realizability Assumption, we further establish the Generalizability Theorem for our proposed DEnD:

**Theorem 4.3** *Given the hypothesis space $\mathcal{H}^*$ with finite Natarajan dimension, the DEnD framework is generalizable in $\mathcal{D}_{XT}^{\mu,b}$ for $\mathcal{H}^*$.*

The proof is provided in Appendix C.5. This theorem highlights DEnD's ability to leverage the differential energy score in its design, ensuring generalizability in theoretical settings. From a practical perspective, as mentioned in the section 5, our method achieves excellent generalization capabilities, which align consistently with our theoretical analysis. Both aspects conclusively highlight the generalizability of our approach.

# 5 Experiments

In this section, we conduct a series of experiments to evaluate generated image detectors within practical scenarios that involve unknown generative models. The experimental results demonstrate that our approach holds significant advantages. (Ablation Studies can be found in Appendix I).

## 5.1 Setup

**Datasets.** We evaluate the performance of generated image detectors on two commonly used datasets: ImageNet [9] and LSUN-BEDROOM [74]. For ImageNet, the generated images are generated

Table 3: Performance (%) of various detectors on GenImage. All training-based methods were trained on images generated by SD V1.4.

| | | | | Models | | | | |
|---|---|---|---|---|---|---|---|---|
| Methods | Midjourney | SD V1.5 | ADM | GLIDE | Wukong | VQDM | BigGAN | Avg ACC(%) |
| | | | | | Training-based Methods | | | |
| ResNet-50 | 54.90 | 99.70 | 53.50 | 61.90 | 98.20 | 56.60 | 52.00 | 68.11 |
| DeiT-S | 55.60 | 99.80 | 49.80 | 58.10 | 98.90 | 56.90 | 53.50 | 67.51 |
| Swin-T | 62.10 | 99.80 | 49.80 | 67.60 | 99.10 | 62.30 | 57.60 | 71.19 |
| CNNspot | 52.80 | 95.90 | 50.10 | 39.80 | 78.60 | 53.40 | 46.80 | 58.63 |
| Spec | 52.00 | 99.20 | 49.70 | 49.80 | 94.80 | 55.60 | 49.80 | 64.41 |
| F3Net | 50.10 | **99.90** | 49.90 | 50.00 | **99.90** | 49.90 | 49.90 | 64.22 |
| GramNet | 54.20 | 99.10 | 50.30 | 54.60 | 98.90 | 50.80 | 51.70 | 65.66 |
| DIRE | 60.20 | 99.80 | 50.90 | 55.00 | 99.20 | 50.10 | 50.20 | 66.49 |
| UnivFD | 73.20 | 84.00 | 55.20 | 76.90 | 75.60 | 56.90 | 80.30 | 71.73 |
| LaRE | 66.40 | 87.10 | 66.70 | 81.30 | 85.50 | 84.40 | 74.00 | 77.91 |
| DRCT | **94.63** | 99.82 | 61.78 | 65.92 | 99.91 | 74.88 | 58.81 | 79.39 |
| GenDet | 89.60 | 96.10 | 58.00 | 78.04 | 92.80 | 66.50 | 75.00 | 79.49 |
| | | | | | Training-free Methods | | | |
| AEROBLADE | 80.30 | 86.89 | 67.20 | 81.57 | 83.74 | 51.10 | 52.53 | 71.90 |
| RIGID | 82.07 | 68.53 | 73.33 | 86.23 | 68.80 | 80.63 | 93.13 | 78.96 |
| DEnD (ours) | 89.44 | 71.88 | **94.46** | **99.07** | 76.27 | **96.61** | **97.84** | **89.37** |

Table 4: The performance of various detectors on Sora.

| | CNNspot | | UnivFD | | NPR | | Methods DRCT | | AEROBLADE | | RIGID | | DEnD (ours) | |
|---|---|---|---|---|---|---|---|---|---|---|---|---|---|---|
| Models | AUROC | AP | AUROC | AP | AUROC | AP | AUROC | AP | AUROC | AP | AUROC | AP | AUROC | AP |
| Sora | 52.85 | 53.29 | 77.06 | 80.69 | 51.92 | 50.25 | 82.53 | 82.28 | 57.13 | 58.00 | 84.22 | 81.98 | **87.35** | **90.57** |
| Open Sora | 50.14 | 51.38 | 67.05 | 68.67 | 50.25 | 51.84 | 81.79 | 80.11 | 55.79 | 62.37 | 73.12 | 75.56 | **90.79** | **93.40** |
| Average | 51.50 | 52.84 | 72.06 | 74.68 | 51.09 | 51.05 | 82.16 | 81.20 | 56.46 | 60.19 | 78.67 | 78.77 | **89.07** | **91.99** |

with ADM [10], ADM-G, LDM [56], DiT-XL2 [51], BigGAN [2], GigaGAN [24], StyleGAN [26], RQ-Transformer [30], and MaskGIT [5]. For LSUN-BEDROOM, generated images are generated with ADM, DDPM [22], iDDPM [44], Diffusion Projected GAN [70], Projected GAN [70], Style-GAN [26] and Unleashing Transformer [1]. To demonstrate the superiority of our method in more realistic scenarios involving unknown generative models, we evaluate the detectors on two general and comprehensive benchmarks : GenImage [79] and AIGCDetectBenchmark [77]. GenImage includes Stable Diffusion V1.4 [56], Stable Diffusion V1.5 [56], GLIDE [45], VQDM [17], Wukong [72], Big-GAN, ADM, and Midjourney [41]. AIGCDetectBenchmark [77] includes ProGAN [25], StyleGAN, BigGAN, StarGAN [7], GauGAN [49], StyleGAN2 [27], WFIR [71], ADM, Glide, Midjourney, Stable Diffusion V1.4, Stable Diffusion V1.5, VQDM, Wukong, DALL-E2 [54]. To demonstrate the generalizability of our method on unavailable generative models, we also evaluate detectors on Sora [47]. The details and sources of the datasets can be found in Appendix G.

**Evaluation Metrics.** We follow in the footsteps of pioneering researchers and adopt the Average Precision (AP) and the Area Under the Receiver Operating Characteristic Curve (AUROC) as our key evaluation metrics. In certain experiments, to ensure comparability with established baselines, we also include accuracy (ACC) as an additional evaluation metric.

**Baselines.** We utilize both training-based and training-free methods as baselines. For training-based methods, we take DIRE [69], CNNspot [68], UnivFD [46], DRCT [6], and NPR [64] as baselines. For some baselines, we get the results reported in their papers, including Frank [14], Durall [11], Patchfor [4], F3Net [52], SelfBland [60], GANDetection [38], LGrad [65], ResNet-50 [19], DeiT-S [66], Swin-T [35], Spec [75],FreDect [13], Fusing [23], LNP [32], GenDet [78], LaRE² [37], and GramNet [36]. For training-free methods, we take AEROBLADE [55] and RIGID [20] as baselines.

**Experimental Details.** In our experiments, we employ the powerful pre-trained self-supervised model DINOv2 [48]. We adopted the DINOv2 ViT-L/14 model, recognized for its optimal balance between speed and performance. We set the batch size $N = 128$ and temperature coefficient $\tau = 0.6$, which show the best performance (see Appendix I.1). Regarding the selection of $m(\mathbf{x})$, we employ Gaussian noise with a mean of 0 and a variance of 0.04 (see Appendix H).

Table 5: Performance (%) on AIGCDetectBenchmark. All training-based methods were trained on images generated by ProGAN.

| Methods | Models | | | | | | | | | | | | | | | |
|---|---|---|---|---|---|---|---|---|---|---|---|---|---|---|---|---|
| | ProGAN | StyleGAN | BigGAN | StarGAN | GauGAN | StyleGAN2 | WFIR | ADM | Glide | Midjourney | SDv1.4 | SDv1.5 | VQDM | Wukong | DALLE2 | Avg ACC(%) |
| CNNSpot | **100.00** | 90.17 | 71.17 | 94.60 | 81.42 | 86.91 | 91.65 | 60.39 | 58.07 | 51.39 | 50.57 | 50.53 | 56.46 | 51.03 | 50.45 | 70.78 |
| FreDect | 99.36 | 78.02 | 81.97 | 94.62 | 80.57 | 66.19 | 50.75 | 63.42 | 54.13 | 45.87 | 38.79 | 39.21 | 77.80 | 40.30 | 34.70 | 64.03 |
| Fusing | **100.00** | 85.20 | 77.40 | 97.00 | 77.00 | 83.30 | 66.80 | 49.00 | 57.20 | 52.20 | 51.00 | 51.40 | 55.10 | 51.70 | 52.80 | 68.38 |
| LNP | 99.67 | **91.75** | 77.75 | **99.92** | 75.39 | **94.64** | 70.85 | 84.73 | 80.52 | 65.55 | **85.55** | **85.67** | 74.46 | **82.06** | 88.75 | 83.84 |
| LGrad | 99.83 | 91.08 | 85.62 | 99.27 | 78.46 | 85.32 | 55.70 | 67.15 | 66.11 | 65.35 | 63.02 | 63.67 | 72.99 | 59.55 | 65.45 | 75.34 |
| DIRE | 95.19 | 83.03 | 70.12 | 95.47 | 67.79 | 75.31 | 58.05 | 75.78 | 71.75 | 58.01 | 49.74 | 49.83 | 53.68 | 54.46 | 66.48 | 68.68 |
| UnivFD | 99.81 | 84.93 | 95.08 | 95.75 | **99.47** | 74.96 | 86.90 | 66.87 | 62.46 | 56.13 | 63.66 | 63.49 | 85.37 | 70.93 | 50.75 | 78.43 |
| DEnD (ours) | 98.88 | 90.24 | **97.08** | 90.95 | 98.54 | 88.33 | **97.25** | **95.37** | **98.25** | **83.17** | 71.99 | 72.27 | **97.68** | 76.71 | **90.65** | **89.82** |

## 5.2 Main Results

**Comparison with Existing Methods.** As shown in Tables 1 and 2, compared to the training-based methods on LSUN-Bedroom and ImageNet, our method is more generalizable and performs better against most generative models, showing significant improvement at the overall level and reflecting the superiority of our generalizable training-free method. Compared to the training-free methods AEROBLADE and RIGID, our method shows substantial improvement against most of generative models. While RIGID employs a noise-based approach based on empirical observation, we derive our approach from the training objectives of self-supervised models. This foundational perspective enabled us to design a more effective differential energy score, achieving better performance. Furthermore, as shown in Table 3 and Table 5, when faced with more advanced and complex generative models, training-based methods generally perform poorly against generative models that were not seen during the training process. In contrast, our method has excellent generalization capabilities and performs significantly better than existing training-based methods. However, due to limitations in the pre-trained model's representational capabilities, our method experiences performance degradation when dealing with certain high-fidelity images, notably those from Stable Diffusion. We posit that this limitation could be alleviated by adopting models with enhanced representational capabilities.

**Discussion on Generalization Capabilities.** Our method demonstrates significant improvements in generalization performance. From a training perspective, conventional training-based approaches often suffer from overfitting issues. As shown in Table 5, models trained on ProGAN exhibit satisfactory performance only when tested on other GAN-generated samples. In contrast, our training-free method inherently avoids overfitting risks. From a theoretical perspective, given the variability of different generative architectures, the patterns of generated images can be highly diverse. This complexity presents significant challenges for training-based methods. In contrast, our method regards the patterns of all generated images as OOD data, maintaining strong generalization capabilities across various generative models. Furthermore, our method provides theoretical guarantees of generalizability, a distinctive advantage absent in existing approaches.

**Evaluation on Sora.** Sora and other video generative models are often of unknown architectures, making the detection of these novel and unknown models more challenging. As demonstrated in Table 4, our experiments on Sora reveal that our approach achieves strong generalization capabilities, attaining competitive performance even when tested on generative models with unknown architectures—a critical advantage absent in existing methods.

## 5.3 Robustness Evaluation

In practical scenarios, detectors are frequently confronted with degraded images. For example, lossy compression may induce artifacts, and noise is typically generated during transmission over communication channels. Following previous works [55], we evaluate the robustness of our detector against such prevalent conditions, encompassing assessments of JPEG compression, Gaussian noise, and Gaussian blur. These experiments are conducted on the ImageNet dataset.

As shown in Figure 3, DEnD demonstrates superior performance across various types of image degradation, reflecting strong robustness. In contrast, other training-based methods often show unsatisfactory performance. This advantage can be credited to the inherent generalization capabilities of our approach, which are underpinned by a solid theoretical foundation, allowing it to consistently classify degraded ID data (natural images) as ID data (natural images).

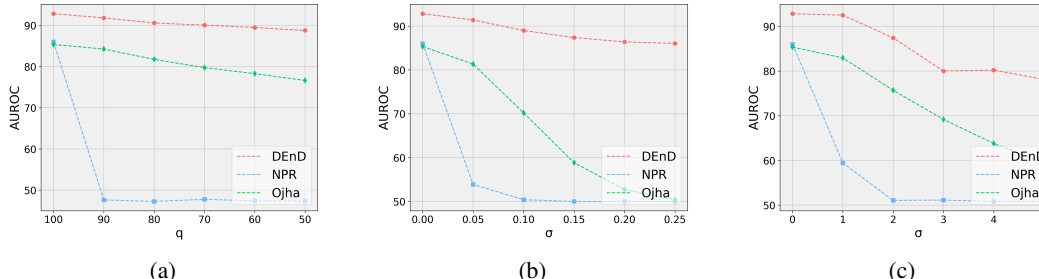

Figure 3: The performance of detectors when faced with degraded images. (a): JPEG with quality $q$. (b): Gaussain noise with standard deviation $\sigma$. (c): Gaussain blur with standard deviation $\sigma$.

## 6 Limitations

**1)** In this work, we formulate generated image detection as an OOD detection task and propose a novel framework inspired by energy-based OOD detection. While our current approach prioritizes energy-based OOD detection, we explicitly acknowledge the potential viability of alternative advanced OOD detection strategies. Future work will focus on exploring the applicability of other OOD detection strategies. **2)** Extensive experiments demonstrate that our method achieves superior performance, which is attributed to our approach with theoretical guarantees. Nevertheless, limited by the training set scope, pre-trained models often fail to realize the full potential of our framework, especially when encountering a real data distribution shift (See Appendix J). In future work, we will attempt to fine-tune the model to attain improved generalizability.

## 7 Conclusion

In this paper, drawing inspiration from the human cognitive ability to discern generated images, we propose a novel perspective on understanding and improving generated image detection: formulating it as an OOD detection task. On this basis, we elucidate the feasibility of employing models trained entirely on natural images for generated image detection. To operationalize this insight, we introduce Differential Energy-based Detection (DEnD), a training-free and generalizable framework for generated image detection. Extensive experiments demonstrate that our approach excels on common benchmarks. Moreover, our method exhibits excellent generalization capabilities, effectively handling generative models with unknown architectures, such as Sora. More broadly, our work not only contributes theoretically but also provides a generated image detection method with superior effectiveness and generalization capabilities, addressing the growing crisis of image forgery.

## Acknowledgments and Disclosure of Funding

This work was supported by NSFC No. 62222117. YGZ was funded by Inno HK Generative AI R&D Center. CW was supported by Zhejiang Provincial Key Research and Development Project (2023C01043), Zhejiang Province Leading Geese Plan (2025C02025), and Academy Of Social Governance Zhejiang University. JY was supported by NSFC No. 62125201, U24B20174.

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

## A    The Discriminative Capacity of Self-Supervised Models

Previous work [61] highlights that evaluation metrics for generative models, such as those based on Inception-V3 [63]—a model trained via supervised learning in label space—primarily focus on category-related semantic information but exhibit insensitivity to features like texture and shape. This limitation also explains why common OOD detection methods trained in label space underperform. [61] emphasizes that self-supervised models, particularly DINOv2, trained on large-scale datasets capture representation spaces that better reflect global, label-agnostic semantic differences between generated and natural images. When designing generative model evaluation metrics, replacing Inception-V3 with self-supervised models like DINOv2 aligns with human evaluation. This observation underscores DINOv2's capability to discern pattern-level discrepancies between natural and generated images. However, directly applying self-supervised model's representation spaces for generated image detection is infeasible (see Figure 4). Compared to evaluating generative models, generated image detection tasks demand higher discriminative power from models. Therefore, building upon the representation spaces learned by self-supervised models, we must further develop algorithmic approaches (our DEnD framework) to achieve effective generated image detection.

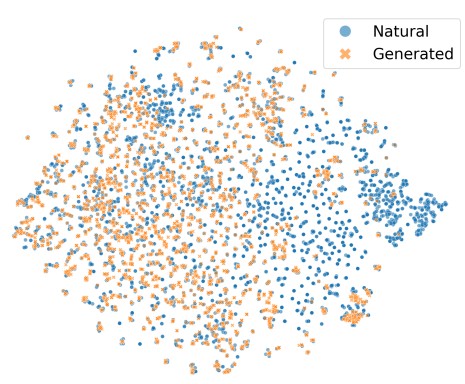

Figure 4: t-SNE visualization of features extracted by DINOv2.

## B    Details of Figure 1

For Figure 1b and Figure 1c, the natural images are from ImageNet and the generated images are generated by ADM. To facilitate visualization and comparison, we normalize the x-axis values to the range $[0, 1]$. For the y-axis, increased density within the distribution is positively correlated with a higher incidence rate of frequency.

## C    Details of The Generalizability of Detectors

### C.1    Generalizability of the Detector.

We set $\mathcal{T}_{\text{all}} = \mathcal{T}_n \cup \mathcal{T}_g$. Given a loss function $\ell : \mathcal{T}_{\text{all}} \times \mathcal{T}_{\text{all}} \to \mathbb{R}_{\geq 0}$ satisfying that $\ell(t_1, t_2) = 0$ if and only if $t_1 = t_2$ and any $h \in \mathcal{H}$, then the risk with respect to $D_{XT}$ is:

$$R_D(h) := \mathbb{E}_{(\mathbf{x}, t) \sim D_{XT}} \ell(h(\mathbf{x}), t). \tag{15}$$

The $\alpha$-risk is:

$$R_D^\alpha(h) := (1 - \alpha) R_D^n(h) + \alpha R_D^g(h), \forall \alpha \in [0, 1], \tag{16}$$

where $R_D^n(h) := \mathbb{E}_{\mathbf{x} \sim D_{X_n}} \ell(h(\mathbf{x}), 1)$, and $R_D^g(h) := \mathbb{E}_{\mathbf{x} \sim D_{X_g}} \ell(h(\mathbf{x}), 2)$. Following the definition of learnability in OOD detection [12], we define the generalizability of the detector as follows:

**Definition C.1** *Given a domain space $\mathcal{D}_{XT}$ and a hypothesis space $\mathcal{H} \subset \{h : \mathcal{X} \to \mathcal{T}_{\text{all}}\}$, we say the generated images detector is **generalizable** in $\mathcal{D}_{XT}$ for $\mathcal{H}$, if there exists an algorithm $\mathbf{A} : \cup_{n'=1}^{+\infty} (\mathcal{X} \times \mathcal{T})^{n'} \to \mathcal{H}$ and a monotonically decreasing sequence $\epsilon_{\text{cons}}(n')$, such that $\epsilon_{\text{cons}}(n') \to 0$, as $n' \to +\infty$, and for any domain $D_{XT} \in \mathcal{D}_{XT}$,*

$$\mathbb{E}_{S \sim D_{X_n T_n}^{n'}} \left[ R_D(\mathbf{A}(S)) - \inf_{h \in \mathcal{H}} R_D(h) \right] \leq \epsilon_{\text{cons}}(n'). \tag{17}$$

*And we say it is **strong generalizable** when the following equation holds for $\forall \alpha \in [0, 1]$ :*

$$\mathbb{E}_{S \sim D_{X_n T_n}^{n'}} \left[ R_D^\alpha(\mathbf{A}(S)) - \inf_{h \in \mathcal{H}} R_D^\alpha(h) \right] \leq \epsilon_{\text{cons}}(n'). \tag{18}$$

In the real world, the distribution of natural images and the distribution of generated images are unknown given that $\pi^{\text{out}}$ can be any value in $[0, 1)$. Therefore, strong generalizability is more aligned with real-world scenarios. In [12], Lemma C.2 indicates that generalizability and strong generalizability are equivalent in certain spaces. Our discussion primarily focuses on these spaces.

**Lemma C.2** *If for any domain $D_{XT} \in \mathcal{D}_{XT}$ and any $\alpha \in [0, 1)$ we have:*

$$D_{XT}^{\alpha} := (1 - \alpha)D_{X_n T_n} + \alpha D_{X_g T_g} \in \mathcal{D}_{XT}, \tag{19}$$

*then Equation 17 and Equation 18 are equivalent in domain space $\mathcal{D}_{XT}$.*

## C.2 Density-based Space.

**Definition C.3** *For any $D_{XT} \in \mathcal{D}_{XT}^{\mu,b}$, there exists a density function $f$ with $1/b \leq f \leq b$ in $supp\mu$ and $0.5 * D_{X_n} + 0.5 * D_{X_g} = \int f \mathrm{d}\mu$, where $\mu$ is a measure defined over $\mathcal{X}$.*

## C.3 Realizability Assumption

**Assumption C.4** *A domain space $\mathcal{D}_{XT}$ and hypothesis space $\mathcal{H}$ satisfy the Realizability Assumption, if for each domain $D_{XT} \in \mathcal{D}_{XT}$, there exists at least one hypothesis function $h^* \in \mathcal{H}$ such that $R_D(h^*) = 0$.*

## C.4 Proof of Theorem 4.2

We prove that there exists $h^* \in \mathcal{H}^*$ such that $R_D(h^*) = 0$ for any $D_{XT} \in \mathcal{D}_{XT}^{\mu,b}$, if there exists $\gamma' \in \mathbb{R}$ such that:

$$\sup_{\mathbf{x} \in \text{supp}(D_{X_n})} \lambda(\mathbf{x}; f, m) < \gamma' < \inf_{\mathbf{x} \in \text{supp}(D_{X_g})} \lambda(\mathbf{x}; f, m). \tag{20}$$

We define $h^* = h_{\gamma'} \in \mathcal{H}^*$:

$$h_{\gamma'}(\mathbf{x}) = \begin{cases} 1 & \text{if } \lambda(\mathbf{x}; f, m) \leq \gamma'. \\ 2 & \text{if } \lambda(\mathbf{x}; f, m) > \gamma'. \end{cases} \tag{21}$$

The risk is:

$$R_D(h^*) = (1 - \pi^{\text{out}})\mathbb{E}_{\mathbf{x} \sim D_{X_n}}\ell(h^*(\mathbf{x}), 1) + \pi^{\text{out}}\mathbb{E}_{\mathbf{x} \sim D_{X_g}}\ell(h^*(\mathbf{x}), 2). \tag{22}$$

For $\mathbf{x} \sim D_{X_n}$, $\lambda(\mathbf{x}; f, m) < \gamma'$, so $h^*(\mathbf{x}) = 1$, hence $\mathbb{E}_{(\mathbf{x}) \sim D_{X_n}}\ell(h^*(\mathbf{x}), 1) = 0$. For $\mathbf{x} \sim D_{X_g}$, $\lambda(\mathbf{x}; f, m) > \gamma'$, so $h^*(\mathbf{x}) = 2$, hence $\mathbb{E}_{\mathbf{x} \sim D_{X_g}}\ell(h^*(\mathbf{x}), 2) = 0$. Thus:

$$R_D(h^*) = (1 - \pi^{\text{out}}) \cdot 0 + \pi^{\text{out}} \cdot 0 = 0. \tag{23}$$

We have completed this proof.

## C.5 Proof of Theorem 4.3

For the model $f$ sufficiently trained on ID data, according to Equation 10 we can obtain that for any $\epsilon > 0$ and for any ID data $\mathbf{x}$:

$$\lambda(\mathbf{x}; f, m) < \epsilon. \tag{24}$$

Therefore, we posit that under ideal conditions the sufficiently trained model satisfies:

$$\sup_{\mathbf{x} \in \text{supp}D_{X_n}} \lambda(\mathbf{x}; f, m) < \gamma' < \inf_{\mathbf{x} \in \text{supp}D_{X_g}} \lambda(\mathbf{x}; f, m). \tag{25}$$

From Theorem 4.2, we can deduce that the Realizability Assumption holds in the DEnD framework.

Hence, by Lemma 4.1, we can deduce that in the density-based space $\mathcal{D}_{XT}^{\mu,b}$, if $\mu(\mathcal{X}) < +\infty$, and the DEnD hypothesis space $\mathcal{H}^*$ has finite Natarajan dimension, then the score-based detector within the DEnD framework is generalizable, which is Theorem 4.3.

## D  Derivation from Equation 8 to Equation 9

Since the feature extractor $f(\mathbf{x})$ is normalized ($\|f(\mathbf{x})\| = 1$), the energy function is bounded:

$$E(\mathbf{x}; f) = \sum_{i=0}^{N} e^{(f(\mathbf{x}), f(\mathbf{x_i}))/\tau} \leq (N+1)e^{1/\tau} = B. \tag{26}$$

That is to say:

$$E(\mathbf{x}; f) \leq B, \quad E(m(\mathbf{x}); f) \leq B. \tag{27}$$

Therefore:

$$|E(\mathbf{x}; f) - E(m(\mathbf{x}); f)| \leq 2B. \tag{28}$$

The training objective in Equation 8 can be expressed as:

$$\min_{\theta} \mathbb{E}_{\mathbf{x} \sim P_{\text{ID}}} \big[ \frac{1}{k} \sum_{i}^{k} E_{m_i}(\mathbf{x}; f_\theta) \big]. \tag{29}$$

After sufficient training, the model ensures that:

$$\mathbb{E}_{\mathbf{x} \sim P_{\text{ID}}} \left[ \frac{1}{k} \sum_{i}^{k} E_{m_i}(\mathbf{x}; f) \right] \leq B - \delta, \tag{30}$$

where $\delta > 0$. Therefore:

$$\mathbb{E}_{\mathbf{x} \sim P_{\text{ID}}} \frac{1}{k} \sum_{i}^{k} [|E(\mathbf{x}; f) - E(m_i(\mathbf{x}); f)|] \leq 2(B - \delta). \tag{31}$$

Since the optimization objective uniformly applies to all $\mathbf{x} \sim P_{\text{ID}}$, the above inequality holds for all natural images. Thus, for any $\epsilon > 0$, choosing $\delta = B - \epsilon/2$ ensures:

$$\frac{1}{k} \sum_{i}^{k} [|E(\mathbf{x}; f) - E(m_i(\mathbf{x}); f)|] \leq \epsilon \quad \forall \mathbf{x} \sim P_{\text{ID}}. \tag{32}$$

That is, Equation 9.

Table 6: Comparison on detecting with different scores.

| Datasets | DEnD (ours) | | Scores $E(\mathbf{x})$ | | $E(m(\mathbf{x}))$ | |
|---|---|---|---|---|---|---|
| | AUROC | AP | AUROC | AP | AUROC | AP |
| ImageNet | **94.81** | **93.33** | 66.38 | 58.76 | 66.78 | 59.64 |

Although our results (Equation 30 and Equation 31) indicate that both $E(\mathbf{x})$ and $E(m(\mathbf{x}))$ can be minimized during training, the close proximity between the distributions of natural images and generated images leads our experiments (see Table 6) to demonstrate that simply reducing the energy of ID data (natural images) to $B - \delta$ is insufficient. During the training process, self-supervised models not only minimize the energy of ID data but also enforce the proximity between $E(\mathbf{x})$ and $E(m(\mathbf{x}))$. From this insight, we define the differential energy score, which imposes comprehensive requirements on both $E(\mathbf{x})$ and $E(m(\mathbf{x}))$, thereby serving as a more discriminative score. Theoretically, the differential energy score on ID data can be minimized to $2B - 2\delta$. Consequently, the self-supervised model outputs lower differential energy scores for natural images while yielding higher scores for generated images. Experimental results demonstrate that the differential energy score achieves superior performance.

Table 7: Comparison on detecting with different OOD detection approaches.

| Datasets | OOD detection approaches | | | | | | | |
|---|---|---|---|---|---|---|---|---|
| | DEnD (ours) | | Energy | | ViM | | KNN | |
| | AUROC | AP | AUROC | AP | AUROC | AP | AUROC | AP |
| ImageNet | **94.81** | **93.33** | 46.36 | 55.12 | 60.38 | 66.11 | 52.12 | 55.95 |

## E  Discussions on Additional OOD Detection Approaches

In Sec. 4.3, we validate and explain the limitations of directly applying the energy-based OOD detection method. By grounding our approach in the training objective of self-supervised learning, we redefine the energy score and introduce our differential energy score. While Sec. 4.2 focuses on one specific OOD detection method, we do not dismiss the potential effectiveness of other approaches. We further evaluate other advanced OOD detection methods, such as KNN [62] and ViM [67]. All methods are trained on LSUN-Bedroom and tested on ImageNet. Experimental results in Table 7 demonstrate that directly applying OOD detection methods designed for label spaces is infeasible for generated image detection. This is because the distinction between natural and generated images lies in high-level patterns rather than simple semantic label differences. Therefore, we adopt category-agnostic self-supervised models and refine the energy score to achieve superior detection performance.

## F  Details of The Threshold

In our experiments, we directly computed the differential energy scores between natural and generated data separately, employing AUROC and AP as evaluation metrics. The assessment process does not involve threshold selection. To clarify our methodology, Equation 12 explicitly adopts the discrimination threshold formulation.

For experiments requiring accuracy-based evaluation, it is quite difficult to manually determine a suitable threshold through visual observation for two large datasets. To find an optimal threshold, we randomly separated 2,000 natural and generated images as a validation set (with no overlap with the test set) and used an algorithm to identify the optimal threshold in the validation set. This threshold was then applied to calculate the accuracy on the test set.

## G  Details of the Datasets

**IMAGENET.** The source of the dataset can be found at `https://github.com/layer6ai-labs/dgm-eval`. We resize the image to $224 \times 224$ resolution as input. The real images are provided by [9]. The generated images include:

- ADM, FID = 11.84.
- ADMG, FID = 5.58.
- BigGAN, FID = 7.94.
- DiT-XL-2, FID = 2.80.
- GigaGAN, FID=4.16.
- LDM, FID=4.29.
- StyleGAN-XL, FID=2.91.
- RQ-Transformer, FID=9.71.
- Mask-GIT, FID=5.63.

**LSUN-BEDROOM.** The source of the dataset can be found at `https://github.com/layer6ai-labs/dgm-eval`. We resize the image to $224 \times 224$ resolution as input. The real images are provided by [74]. The generated images include:

- ADM, FID=2.20.
- DDPM, FID=5.18.
- iDDPM, FID=4.54.
- StyleGAN, FID=2.65.
- Diffusion-Projected GAN, FID=1.79.
- Projected GAN, FID=2.23.
- Unleashing Transformers, FID=3.58.

**GenImage.** The source of the dataset can be found at `https://github.com/GenImage-Dataset/GenImage`. We resize the image to $224 \times 224$ resolution as input. The images are provided by [79]. The generated images include:

- Midjourney.
- SD V1.4.
- SD V1.5.
- ADM.
- GLIDE.
- Wukong.
- VQDM.
- BigGAN.

**Sora.** To demonstrate the generalizability of our method on generative models with unknown architectures, we collect Sora [47] generated videos and sample them to obtain images. We collect the official demonstration videos and extract frames to obtain 5,000 images. Additionally, we utilize the open-source OpenSora [76] project to generate 100 videos, from which we also extract frames to get another 5,000 images. We use these images as generated images, and we randomly select 5,000 images from LAION [57] as natural images. We resize the images to $224 \times 224$ resolution. We employ these images to evaluate the generalizability of our method and compare them with baselines.

## H   Random Transformation Distribution

Table 8: DEnD's performances on datasets with different transformations.

| Datasets | Transformations | | | | | | | |
| | Gaussian filter | | Gaussian noise | | random rotate | | salt and pepper noise | |
| | AUROC | AP | AUROC | AP | AUROC | AP | AUROC | AP |
| --- | --- | --- | --- | --- | --- | --- | --- | --- |
| ImageNet | 92.82 | 91.09 | **94.81** | **93.33** | 88.57 | 90.04 | 92.12 | 91.15 |
| LSUN | 94.54 | 93.11 | **96.05** | **95.03** | 64.21 | 60.06 | 92.10 | 90.07 |
| Genimage | 79.85 | 76.96 | **93.04** | **90.32** | 89.29 | 85.32 | 89.57 | 85.33 |

In our experiments, we evaluate diverse random transformation strategies $m(\mathbf{x})$ used to generate positive samples during self-supervised model training, including adding Gaussian noise, applying Gaussian filter, adding salt-and-pepper noise, and rotating at random angles. Each of these transformations demonstrated robust performance, indicating their effectiveness in our study.

Our approach is highly adaptable, demonstrating commendable effectiveness with a variety of common random transformations. To complement our findings, we compared the average performance of some common image transformations across these datasets (see Table 8). In our Gaussian filtering, we set the $\sigma = 0.7$. In the Gaussian noise, we set the $mean = 0, std = 0.04$. In random rotation, we set the rotation angle to randomly select between $(-10°, 10°)$. In salt and pepper noise, we set the probability of adding salt noise and pepper noise to both be $10^{-4}$. Overall, adding Gaussian noise shows superior average performance. Consequently, we employed Gaussian noise in our experiments.

# I  Ablation Studies

## I.1  Temperature Coefficient and Batch Size Selection

Table 9: DEnD's performance across varying temperature coefficients.

| Datasets | Temperature coefficient | | | | | | | |
|---|---|---|---|---|---|---|---|---|
| | $t = 0.4$ | | $t = 0.6$ | | $t = 1.0$ | | $t = 5.0$ | |
| | AUROC | AP | AUROC | AP | AUROC | AP | AUROC | AP |
| ImageNet | 93.33 | 92.10 | **94.81** | **93.33** | 94.41 | 92.77 | 94.52 | 92.80 |

Regarding the temperature coefficient, our experiments (see Table 9) demonstrate that it has minimal impact on the results. Specifically, different values within a reasonable range do not lead to significant performance variations.

Table 10: DEnD's performance across varying batch sizes.

| Datasets | Batch sizes | | | | | | | |
|---|---|---|---|---|---|---|---|---|
| | $N = 16$ | | $N = 64$ | | $N = 128$ | | $N = 256$ | |
| | AUROC | AP | AUROC | AP | AUROC | AP | AUROC | AP |
| ImageNet | 89.53 | 86.22 | 93.30 | 92.81 | **94.81** | **93.33** | 94.11 | 92.49 |

As for batch size selection, our experiment (see Table 10) reveals that larger values of N generally improve effectiveness but also increase computational complexity. To balance accuracy and computational efficiency, we ultimately adopted N=128 in our experiments.

## I.2  The Selection of Self-supervised Models

Table 11: Comparison on detecting with different self-supervised models.

| Datasets | Self-supervised models | | | | | |
|---|---|---|---|---|---|---|
| | DINO | | CLIP | | DINOv2 | |
| | AUROC | AP | AUROC | AP | AUROC | AP |
| ImageNet | 69.21 | 66.87 | 75.42 | 80.21 | **94.81** | **93.33** |

We also experimented with self-supervised models, such as DINO [3] and CLIP [53]. The results (see Table 11) demonstrate that other self-supervised models significantly underperform compared to DINOv2. Our method relies not only on the discriminative power of our differential energy score but also on the representational capability of the adopted self-supervised model. Unlike DINO and CLIP, DINOv2 better captures global pattern-level differences. These results align with [61], which states that DINOv2 usually focuses on the image structure as a whole while still identifying objects of importance—a capability lacking in other self-supervised models. Consequently, [61] employs DINOv2's representational space for generated image evaluation. Similarly, to better characterize pattern-level discrepancies between natural and generated images, our DEnD framework adopts a pre-trained DINOv2 model.

# J  Further Discussion

Our method operates on the premise that the self-supervised model is pre-trained on an extensive and diverse corpus of real images, thus ensuring lower differential energy scores for all natural images (ID data). In practice, we employ DINOv2 ViT-L/14, pre-trained on the LVD-142M [48] dataset, which is designed to cover as many natural image domains as possible. Although extensive experiments confirm our method's outstanding performance, and its theoretical foundation—regarding the patterns of all generated images as OOD data—enables generalization to unseen architectures, we observe

Table 12: The performance of detectors under real data distribution shift. We investigate the impact of different real data sources, keeping the generated images sources consistent with Table 1.

| Real Data | Methods | | | |
| | RIGID | | DEnD (ours) | |
| | AUROC | AP | AUROC | AP |
| --- | --- | --- | --- | --- |
| ImageNet | 83.73 | 81.69 | **94.81** | **93.33** |
| COCO | 67.58 | 65.07 | **89.84** | **88.61** |

a key limitation. As shown in Table 12, when we evaluated our method using COCO [31] as the source of natural images—a dataset exhibiting a potential distribution shift from the training set of DINOv2—we observed a noticeable performance degradation. This observation is consistent with our theoretical framework, as the self-supervised model is optimized to minimize the differential energy score only for real data from distributions encountered during training. We also observed that RIGID, another training-free method that also leverages DINOv2, suffers an even more significant performance drop. This underscores the limitations stemming from the scope of DINOv2's training data. As part of our future work, we plan to fine-tune our model on larger and more diverse real-world datasets to further enhance detection performance.

## K    Compute Resources

As a training-free method, our approach exhibits minimal computational overhead, which stands as one of the key advantages of this work. All experiments were conducted on a single NVIDIA GeForce RTX 4090 GPU with 24 GB memory. With a batch size of 128, the inference speed is approximately 0.5 seconds per batch.

## L    Broader Impacts

This article proposes a novel approach for detecting generated images, offering an effective solution to identify and mitigate the proliferation of synthetic media. Our work contributes to reducing the spread of misinformation through synthetic content, with significant potential societal benefits. We recognize that generated images give rise to ethical concerns, particularly regarding privacy protection and consent issues. The proposed method addresses these challenges by establishing a reliable detection framework. This research not only advances the field of generated image detection, but also represents a critical step toward preserving digital media integrity in the AI era.

