# OpenReview forum: "Towards Generalizable Detector for Generated Image"
_NeurIPS.cc/2025/Conference — NeurIPS 2025 poster_

### Official Review · Reviewer_P2GB · 2025-06-29

**Clarity:** 3
**Significance:** 2
**Originality:** 2
**Rating:** 4
**Confidence:** 2

**Summary:**

This paper introduces a framework for detecting generated images by reframing the problem as an Out-of-Distribution (OOD) detection task. The authors propose Differential Energy-based Detection (DEnD), which leverages the differential energy scores computed via a self-supervised pretrained model (DINOv2). The core insight is that natural images (ID data) will have low differential energy scores, while generated images (OOD data) will have high scores. The paper provides theoretical guarantees of generalizability and conducts extensive experiments across several benchmarks (ImageNet, LSUN-BEDROOM, GenImage, AIGCDetectBenchmark, Sora). The results indicate that DEnD achieves state-of-the-art performance among training-free methods and often outperforms training-based detectors.

**Questions:**

- The authors compare DEnD’s performance across different self-supervised models (e.g., DINO, CLIP, DINOv2) and demonstrate DINOv2’s superiority. Given this dependence, could hybrid or ensemble approaches combining multiple self-supervised representations further improve robustness or mitigate reliance on a single model?

- While the paper includes evaluations on various generative models, including some high-fidelity examples (e.g., StyleGAN-XL, Sora), can the authors provide a more focused analysis of cases where generated images are nearly indistinguishable from natural ones (e.g., Stable Diffusion or future models), and quantify how separation degrades in such scenarios?

- Given the method’s reliance on differential energy scoring, how robust is DEnD to adversarial perturbations applied to natural images that could artificially inflate differential energy scores and lead to misclassification?

- The paper presents extensive ablations and robustness tests but does not report statistical significance measures (e.g., error bars, confidence intervals, or variability across seeds). Could the authors clarify whether multiple trials were conducted and, if so, provide formal statistical analyses to support the reported performance?

**Ethical Concerns:**

["NO or VERY MINOR ethics concerns only"]

**Final Justification:**

The authors have successfully addressed my concerns during the rebuttal phase.

Accordingly, I have raised my final rating to 4 (Borderline accept).

However, I am not an expert in the field of generated image detection, which makes it difficult for me to be certain that this score accurately reflects the paper's technical novelty.

For this reason, I have lowered my confidence score to 2.

I hope the Area Chair will make the final decision based on the reviews from experts in the field.

**Limitations:**

See Weaknesses and Questions

**Paper Formatting Concerns:**

No Paper Formatting Concerns

**Quality:**

3

**Strengths And Weaknesses:**

**Strengths:**

- The paper presents a fresh perspective by casting generated image detection as an OOD detection problem.

- DEnD eliminates the need for task-specific training, relying on pretrained self-supervised models. This reduces training costs and mitigates overfitting risks.

- The experiments cover a wide range of generative models, including unseen and unknown architectures (e.g., Sora).

**Weaknesses:**


- The success of DEnD is strongly dependent on the representational capacity of the selected pretrained model, particularly DINOv2. Although the authors provide comparisons with alternative self-supervised models (e.g., CLIP, DINO), these alternatives significantly underperform, confirming the reliance on DINOv2 and limiting flexibility in model choice.

- While the theoretical framing—casting generated image detection as an OOD detection problem—is novel, the proposed method itself represents a relatively straightforward application of differential energy scoring using existing self-supervised representations. The algorithmic contribution beyond the reframing and score definition is incremental.

- The method assumes that generated images consistently exhibit high differential energy relative to natural images. While this holds in many cases, there is no deep analysis or ablation to test this assumption against high-fidelity generative models or adversarially crafted fakes.

- The code is not released at submission, and reproducibility relies heavily on future promises rather than current availability. Important hyperparameter choices (e.g., transformation distributions, thresholds) are not explored in depth.

---

> ### Author Rebuttal · Authors · 2025-07-31
>
> We sincerely appreciate the time and effort you have dedicated to reviewing our paper, as well as your constructive comments. In response to your feedback, we have provided detailed responses below. We hope our responses address your concerns and contribute to enhancing our work.
>
> **W.1 The success of DEnD is strongly dependent on the representational capacity of the selected pretrained model, particularly DINOv2. Given this dependence, could hybrid or ensemble approaches combining multiple self-supervised representations further improve robustness or mitigate reliance on a single model?**
>
> A.1 Our method shows significant performance gain when DINOv2 is utilized. This is consistent with our motivation, namely, our OOD perspective for detection transforms the generalization challenge of generated image detection into the problem of fitting natural image patterns. Thus, DINOv2 showing relatively strong performance when fitting natural distributions can achieve better performance under various settings.
>
> Moreover, we would like to highlight that our method leverages both the representational capacity of DINOv2 and our designed differential energy score. As demonstrated in Appendix A, directly applying DINOv2 for detection yields suboptimal results.
>
> Following your kind suggestion, we conduct experiments incorporating features from multiple self-supervised models (CLIP, DINO, and DINOv2) and fusing them to mitigate the reliance on a single model. Aligning with your insightful comments, our experimental results on ImageNet demonstrate that integrating features from all three models yields superior performance compared to using CLIP or DINO features individually. We have added these results and discussions to our revision. Although performance degrades significantly when using CLIP or DINO alone, fusing features from multiple self-supervised models – as you suggested – can lead to some improvement.
>
> ||DINO|CLIP|DINOv2|fusing|
> |-|-|-|-|-|
> |AUROC|69.21|75.42|94.81|82.13|
>
> **W.2 While the theoretical framing—casting generated image detection as an OOD detection problem—is novel, the proposed method itself represents a relatively straightforward application of differential energy scoring using existing self-supervised representations. The algorithmic contribution beyond the reframing and score definition is incremental.**
>
> A.2 First, our paper’s contributions are not limited to the design of the detection method. We make significant theoretical contributions by modeling generated image detection as an OOD detection task and providing a theoretical explanation for the generalizability of current zero-shot generated image detection methods, which utilize models pretrained entirely on real images to detect generated images. This is something no one has done in the field of generated image detection.
>
> Second, although our method builds upon existing self-supervised and energy-based techniques[1], the energy-based OOD detection work[1] mentioned in our paper only served as an inspiration for our approach. Our differential energy-based detection is fundamentally different from [1].
>
> More than that, our simple method significantly enhances generalization in generated image detection, achieving excellent detection performance even for high-fidelity images. Compared to prior work, our method requires no training, substantially reducing computational overhead while improving generalization. Such a simple yet effective approach can still make significant contributions to the field of generated image detection. For instance, reviewers 9KFU and SCX8 both affirmed the importance of our work in their comments.
>
> **W.3 The method assumes that generated images consistently exhibit high differential energy relative to natural images. While this holds in many cases, there is no deep analysis or ablation to test this assumption against high-fidelity generative models or adversarially crafted fakes.**
>
> A.3 Thank you for your suggestions! We have added discussions on the detection performance for high-fidelity images and an analysis of adversarially crafted fakes. As you pointed out, in a few cases, some highly realistic samples (e.g., those from Stable Diffusion) have scores that are quite close to those of real data, making them difficult to distinguish, which also leads to relatively poorer performance on Stable Diffusion in our experiments. However, overall, for most high-fidelity samples, such as those from Sora, Flux, and Meta Movie Gen, our detector can still output scores with high discriminative power, achieving excellent detection performance. For detailed content, please refer to our responses to Questions 2 and Question 3.
>
> **W.4 The code is not released at submission, and reproducibility relies heavily on future promises rather than current availability. Important hyperparameter choices (e.g., transformation distributions, thresholds) are not explored in depth.**
>
> A.4 We will release the code with an anonymous link.
>
> We have thoroughly discussed and provided the selection of important hyperparameters used in our experiments in Appendices F, H, and I. In response to your valuable comments, we will highlight the details of hyperparameter choices in the main page. As our reported results primarily consist of AUROC and AP, these metrics do not involve threshold calculations. For a more in-depth discussion on thresholds, please refer to our response to Reviewer 9KFU regarding Question 2.
>
> **Q.1 could hybrid or ensemble approaches combining multiple self-supervised representations further improve robustness or mitigate reliance on a single model?**
>
> A.1 Please refer to our response to Weakness 1.
>
> **Q.2 While the paper includes evaluations on various generative models, including some high-fidelity examples (e.g., StyleGAN-XL, Sora), can the authors provide a more focused analysis of cases where generated images are nearly indistinguishable from natural ones (e.g., Stable Diffusion or future models), and quantify how separation degrades in such scenarios.**
>
> A.2 Our experimental results indicate that our detector achieves exceptional performance when handling low-generation-quality models such as BigGAN. For high-fidelity generated images (e.g., from Stable Diffusion and Sora), we do observe performance degradation. Notably, Stable Diffusion presented a significant challenge in our tests. However, this does not imply fundamental limitations in detecting high-quality generated content. As evidenced by our experiments on Sora and supplementary tests(AUROC) on models like FLUX, Meta Movie Gen and ltx-video[2], our method maintains robust detection capabilities even for state-of-the-art generations. In summary, while demonstrating strong performance against most high-quality generated samples, we acknowledge that exceptionally challenging cases like Stable Diffusion can cause noticeable performance reduction. We will prioritize enhancing performance on high-fidelity generative models as a key focus in our future work.
> ||Sora|FLUX|Meta Movie Gen|LTX-Video|
> |-|-|-|-|-|
> |DRCT|82.53|91.10|90.22|94.17|90.22|
> |DEnD(ours)|90.22|92.87|92.41|97.20|92.41|
>
> **Q.3 Given the method’s reliance on differential energy scoring, how robust is DEnD to adversarial perturbations applied to natural images that could artificially inflate differential energy scores and lead to misclassification?**
>
> A.3 Thank you for your suggestions. Currently, most generated image detection studies do not address robustness against adversarial perturbations, which is also a critical metric for evaluating detectors. We experimentally evaluated our detector's performance(AUROC) on ImageNet when adversarial perturbations were applied to real images using FGSM. The results show a significant performance drop when adversarial perturbations are introduced, indicating that maintaining robustness against adversarial perturbations remains a challenge in the field of generated image detection. Thank you for your insightful feedback. We will consider improving robustness against adversarial perturbations as a limitation of our current work and a key focus for future research.
>
> ||epsilon=0.01|epsilon=0.03|epsilon=0.035|
> |-|-|-|-|
> |DRCT|67.04|65.70|65.48|
> |DEnD(ours)|73.91|70.58|65.53|
>
> **Q.4 The paper presents extensive ablations and robustness tests but does not report statistical significance measures (e.g., error bars, confidence intervals, or variability across seeds). Could the authors clarify whether multiple trials were conducted and, if so, provide formal statistical analyses to support the reported performance?**
>
> A.4 Thank you for highlighting the need for statistical significance measures. We have updated the Appendix with these analyses.
> We confirm that all experiments in our paper were conducted with five independent trials, and results were averaged. Due to the exceptional stability of our method (evidenced by extremely low standard deviations) and minimal stochastic influence, we did not originally include variability metrics in the paper. We have now calculated and present the statistical significance measures for experimental results on ImageNet below. The low standard deviations and tight confidence intervals indicate consistent model performance. Although our process involves some randomness in the random transformations, our repeated experiments demonstrate that our results are highly stable and minimally affected by this randomness.
>
> AUROC: Mean = 0.9481, Std = 0.00167, 95% CI = [0.94603, 0.95017]
>
> AP: Mean = 0.9333, Std = 0.00293, 95% CI = [0.92966, 0.93694]
>
> [1] Liu et al., Energy-based Out-of-distribution Detection.
>
> [2] HaCohen, et al., Ltx-video: Realtime video latent diffusion.

---

> > ### Comment · Reviewer_P2GB · 2025-08-05
> >
> > Thank you for your response, which has addressed most of my concerns. Accordingly, I will raise my score to 4.
> >
> > However, I must note that I am not an expert in this field, which makes it challenging for me to confidently assess the paper's technical novelty.
> >
> > For this reason, I will lower my confidence score to 2. I would like the Area Chair to downweight my review in the final decision and to rely on the judgment of expert reviewers.

---

> > > ### Author Response · Authors · 2025-08-05
> > >
> > > Thank you for your constructive feedback and revised evaluation!
> > >
> > > We particularly value your perspective, as one of our goals is to make our work accessible to broader audiences beyond our immediate field. Your comments have been highly valuable and have significantly enhanced the quality of our paper. All points raised in your review have been fully addressed in the revised manuscript.
> > >
> > > Should you have any further questions or concerns, we welcome the opportunity for further discussion. We would be pleased to engage in a deeper exchange of ideas.

---

### Official Review · Reviewer_9kfu · 2025-06-30

**Clarity:** 4
**Significance:** 3
**Originality:** 4
**Rating:** 5
**Confidence:** 3

**Summary:**

This paper frames the problem of AI-generated image detection as an out-of-distribution (OOD) detection problem, where generated images are the out-of-distribution and real (natural) images are in-distribution. Based on this formulation, the paper proposes a novel approach called differential energy-based detection (DEnD) that utilizes a well-known self-supervised model (DINOv2) which only has seen real images in its training phase. DEnD performs AI-generated image detection by calculating the difference of the pre-trained model energy of an input image and it's transformed version ($\lambda$). Based on the theoretical work, real images exhibit lower $\lambda$ values than generated images, making it a reliable representation for discerning real vs AI-generated images. Additionally, the paper provides guarantees of generalizability to unseen generative models through theoretical work, a key element missing in previous works. Finally, the proposed method shows strong performance on a variety of datasets and comparing with state-of-the-art baselines.

**Questions:**

I am willing to increase my score to accept or strong accept based on the authors' response to the following unanswered questions:

1. How does the proposed method perform when there is a real data distributional shift between the training dataset of the pre-trained self-supervised model and real test data. For example, DINOv2 is trained on ImageNet, how does the proposed method perform on a dataset consisting of real images from COCO dataset? Is it going to mistake real images from other datasets as OOD (AI-generated) or is it robust to this distributional shift? If any, how significant is the performance drop in this scenario?
2. How exactly is the threshold selected? What exactly does the validation data mentioned in section F include? How is the optimal threshold calculated? Importantly, how does the accuracy change as a function of the threshold? How sensitive is the accuracy to the validation data?
When comparing with other methods, how the thresholds for those methods are selected?
3. How does the proposed method perform on other types of video-generaton methods, such as LTX-Video [B] or Meta Movie Gen?
4. How robust is the method to resizing, a very common post-processing operation? This should be added to section J.
5. Why the evaluation metrics differ across different experiments? Why are all three metrics not reported in all experiments?
6. How does the proposed method compare to FSD [B]? This is a recently published detection method which similar to this paper do not require AI-generated training data. Since I do not see their code being public, I think at least a high-level comparison in the background section is valuable.

[A] HaCohen, Yoav, et al. "Ltx-video: Realtime video latent diffusion." arXiv preprint arXiv:2501.00103 (2024).

[B] Nguyen, Tai D et al. "Forensic self-descriptions are all you need for zero-shot detection, open-set source attribution, and clustering of ai-generated images." CVPR 2025

**Ethical Concerns:**

["NO or VERY MINOR ethics concerns only"]

**Final Justification:**

The authors addressed all of my concerns in the rebuttal and are adding new experimental results accordingly to their paper (or the appendix). Therefore, I think the paper is in a better shape now and I raise my rating.

**Limitations:**

I think the limitations should be discussed in more detail. For example, what are the real limitations and failure modes of the proposed method? Answers to some of the questions mentioned in the "Questions" section can be added to the "Limitations" to inform the reader of the real limitations of the proposed method.

**Paper Formatting Concerns:**

No major formatting issues were identified.

**Quality:**

3

**Strengths And Weaknesses:**

**Strengths**
1. The paper addresses a timely problem of AI-generated detection through an under-explored perspective and addresses a fundamental limitation of existing works: Ensuring the generalizability of the detectors.
2. The paper is clear, easy to follow and well-written. The flow of information is logical.
3. Theoretical work is sound and ensures the proposed method's generalizability to unseen generative models under mild assumptions.
4. The proposed method for AI-generated image detection is novel and well-motiviated and shows strong performance through experimental validation.

**Weaknesses**
1. Considering the method is "training-free" and utilizes DINOv2, which is trained on ImageNet, it is not clear how the proposed method performs when there is a real data distribution shift at test time.
2. Although an important aspect for adoption the of proposed method to operational environments, the mechanism of selecting a threshold is ambiguous. It is unclear what does the "validation" data in section F include and how the optimal threshold is calculated. Additionally, the process of selecting a threshold for competing methods is not discussed.
For more weaknesses refer to the "Questions" section.

---

> ### Author Rebuttal · Authors · 2025-07-31
>
> We are deeply grateful for your constructive suggestions. To address your insightful comments, we have meticulously prepared point-by-point responses below, all of which have been thoroughly integrated into the revised paper. We hope our responses address your concerns and contribute to enhancing our work.
>
> **W.1 It is not clear how the proposed method performs when there is a real data distribution shift at test time. For example, DINOv2 is trained on ImageNet, how does the proposed method perform on a dataset consisting of real images from COCO dataset?**
>
> A.1 We sincerely appreciate your in-depth and inspiring comments. This is inherently related to the potential performance degradation of our detectors under distribution shift scenarios. Following your valuable suggestion, we conduct supplementary experiments by replacing the ImageNet real data with COCO.
>
> ||ImageNet|COCO|
> |-|-|-|
> |DRCT|90.36|85.13|
> |DEnD(ours)|94.81|89.84|
>
> When there is a distribution shift between the test data and the training data, a decline in performance is inevitable. The results confirm that while our method experiences limited performance degradation when confronted with this distribution shift, it maintains robust detection capabilities overall. Specifically, the AUROC decreased from 94.81% to 89.84% , demonstrating that our approach preserves reasonable robustness under domain shift conditions. This is consistent with your intuition. However, compared to the baseline, our method still achieves better results. In the future, we will continue to explore the impact of distribution shift on generated image detection.
>
> Thanks again for this constructive feedback. Accordingly, we have included comprehensive analysis and discussion of distribution shift scenarios in our revised version.
>
> **W.2 Additionally, the process of selecting a threshold for competing methods is not discussed.
> When comparing with other methods, how the thresholds for those methods are selected?**
>
> A.2 We have added the following details in our experiments.
>
> Regarding the threshold selection for other competing methods, for a significant portion of these methods, we directly adopted the experimental results provided in their papers. For the remaining methods, we used the optimal thresholds discussed in their papers or the default thresholds provided in their open-source code.
>
> **Q.1 How does the proposed method perform when there is a real data distributional shift between the training dataset of the pre-trained self-supervised model and real test data.**
>
> A.1 Please refer to our response to Weakness 1.
>
> **Q.2 The mechanism of selecting a threshold is not ambiguous. It is unclear what does the "validation" data in section F include and how the optimal threshold is calculated. Importantly, how does the accuracy change as a function of the threshold? How sensitive is the accuracy to the validation data?**
>
> A.2 Thanks for pointing out this potentially confusing setting.
>
> In the experiment, it is quite difficult to manually determine a suitable threshold through visual observation for two large datasets. To find an optimal threshold, we randomly separated 2,000 real and generated images as a validation set (with no overlap with the test set) and used an algorithm to identify the optimal threshold in the validation set. This threshold was then applied to calculate the accuracy on the test set.
>
> As for our algorithm to identify the optimal threshold, we merge and sort two classes of data to generate candidate thresholds (midpoints of adjacent points), test both classification directions for each threshold, and select the median of thresholds maximizing correct classifications as the optimal threshold.
>
> Based on your valuable suggestions, we qualitatively analyzed the impact of threshold selection on accuracy using ImageNet, with results shown in the table below. It can be observed that a threshold score of 1.0 yields the highest accuracy on the test set. In our experiments, we used 1.0 as the optimal threshold for calculating accuracy on the test set. Our threshold selection is not sensitive to the choice of validation set; we conducted repeated experiments by randomly selecting validation sets multiple times and calculating the optimal threshold: 0.9403, 1.0125, 1.1845. In our reported results, we adopted 1.0 as the final threshold score.
> We will include the above content in our appendix to avoid potential confusion.
>
> ||th=0.8|th=1.0|th=1.2|th=1.5|th=2.0|
> |-|-|-|-|-|-|
> |ACC|77.14|86.09|85.24|73.92|
>
> **Q.3 How does the proposed method perform on other types of video-generaton methods, such as LTX-Video [1] or Meta Movie Gen?**
>
> A.3 We appreciate your constructive comments. We believe the following experiments and discussions can significantly improve the quality of our work. We have added the following results(AUROC) to our revision.
>
> To further verify the effectiveness of our method, we consider the LTX-Video and Meta Movie Gen models that are highly advanced video generated models. Evaluating on these models would demonstrates the generalization ability of our method. Thus, we conduct experiments on these video generation models using the same experimental settings as for Sora. The results show that our method maintains strong generalization performance when handling these high-quality video generation models, which is a significant advantage of our approach.
>
> ||LTX-Video|Meta Movie Gen|
> |-|-|-|
> |DRCT|90.22|94.17|
> |DEnD(ours)|92.41|97.20|
>
> **Q.4 How robust is the method to resizing, a very common post-processing operation?**
>
> A.4 Thank you for your kind suggestion. Resizing is indeed a common operation in robustness evaluation. In our previous experiments, most images had an initial resolution of 256×256, while datasets such as GENimage and Sora contained images with varying sizes. Due to the requirements of the DINOv2 model, all images were resized to 224×224 before being input to the model. Following your valuable suggestion, we have conducted additional experiments to evaluate the model’s performance when the input images are initially resized to 64×64, 128×128, 256×256, and 512×512. The results have been incorporated into Section J to address your concern.
>
> Overall, our method demonstrates good robustness against resize operations. However, it can be observed that when the original image is resized to 64x64, the detection performance experiences a relatively significant decline. This may result from the fact that a substantial amount of important information in the image is lost during this process, thereby affecting the detector's performance. In contrast, when the image is resized to 512x512, the key information in the image is largely preserved, allowing the detection performance to remain stable.
>
> ||64|128|256|512|
> |-|-|-|-|-|
> |AUROC|86.57|89.16|94.81|93.06|
>
> Thanks again for your valuable feedback. We have added the above results to our revision.
>
> **Q.5 Why the evaluation metrics differ across different experiments? Why are all three metrics not reported in all experiments?**
>
> A.5 Following previous works, we adopt AUROC and AP as the primary evaluation metrics. However, since some of our baselines also report accuracy, we include ACC as an auxiliary metric on certain datasets to remain consistent with these baselines. Due to space constraints, we do not report all three metrics for every experiment. The AUROC, ACC, and AP results of the main experiments presented in the paper are summarized below; following your suggestion, we will provide the complete evaluation metrics for these experiments in the appendix.
>
> ||ImageNet|LSUN|GENImage|
> |-|-|-|-|
> |avg AUROC|94.81|96.05|92.97|
> |avg AP|93.33|95.03|91.05|
> |avg ACC|86.09|86.68|89.37|
>
> **Q.6 How does the proposed method compare to FSD [2]? This is a recently published detection method which similar to this paper do not require AI-generated training data. Since I do not see their code being public, I think at least a high-level comparison in the background section is valuable.**
>
> A.6 FSD[2] is also an outstanding work in the field of generated image detection, and it shares some similarities with our approach, i.e., using models trained on real images for generated image detection. We have included a high-level comparison with this method in the related work section of the final version:
>
> To achieve generated image detection using models trained solely on real images, Nguyen et al. introduces a forensic self-description (FSD) framework that extracts forensic microstructures from images and models the distribution of real images with a Gaussian mixture model, whereas our method formulates generated image detection as an out-of-distribution (OOD) detection task, leveraging a self-supervised visual model pretrained on extensive real data. While our work shares similar goals with FSD, it differs in implementation approach.
>
> **L.1 I think the limitations should be discussed in more detail.**
>
> A.1 We appreciate your kind suggestion. We have refined the content in our limitations section based on the issues you raised.
>
> 1. In this work, we employ the DINOv2 model pre-trained on extensive real-world data. However, when distribution shift exists between test data and training data, noticeable performance degradation occurs despite generally favorable experimental outcomes.
>
> 2. Although our method demonstrates robust performance against high-fidelity generated images (e.g., Sora-generated content), it exhibits suboptimal detection efficacy for certain advanced generative models such as Stable Diffusion.
>
> [1] HaCohen, et al., Ltx-video: Realtime video latent diffusion.
>
> [2] Nguyen et al., Forensic self-descriptions are all you need for zero-shot detection, open-set source attribution, and clustering of ai-generated images.

---

> > ### Comment · Reviewer_9kfu · 2025-08-04
> >
> > I thank the authors for their comprehensive responses. I still have a minor question. Regarding Q.2, can the authors clarify the source or sources of generated images used in the validation set? What I'm wondering here is that whether the proposed method is able to maintain high accuracy even when the threshold is found using data from only one source of generated images.

---

> > > ### Author Response · Authors · 2025-08-05
> > >
> > > Thank you for your prompt response and valuable suggestions! As you rightly noted, we initially did not clarify the sources of generated images in our validation set. Taking ImageNet as an example, the generated images were derived from nine distinct generative models. During validation set construction, we randomly sampled from the entire pool of generated images without limiting to any specific generator type.
> > >
> > > To address your concern regarding *"whether the proposed method maintains high accuracy when thresholds are calibrated using single-source generated data"*, we conducted dedicated experiments:
> > >
> > > 1. To rigorously evaluate the stability of our threshold selection, we systematically computed optimal thresholds for each individual generative model among the nine candidates. The experimental results revealed that all derived thresholds consistently converged within a narrow range centered at 1.0, which aligns with our conclusion in response to Q2 regarding the impact of varying thresholds. This consistency provides empirical evidence of minimal dependency on specific generative sources during threshold determination.
> > >
> > > ||BigGAN|Mask-GIT|RQ-Transformer|ADM|ADMG|DiT-XL-2|GilaGAN|LDM|styleGAN|
> > > |-|-|-|-|-|-|-|-|-|-|
> > > |optimal threshold|1.31|1.12|1.46|1.14|0.98|0.88|1.18|1.01|1.06|
> > >
> > > 2. To quantify generalization performance of the threshold, we fix the threshold at 1.0 and report the accuracy for each generative model, which consistent with the settings in our response to Q2. The experimental results demonstrate that a fixed threshold maintains high accuracy across diverse generative models, confirming that our method's generalization capability is independent of thresholds calibrated on any single generative model.
> > >
> > > ||BigGAN|Mask-GIT|RQ-Transformer|ADM|ADMG|DiT-XL-2|GilaGAN|LDM|styleGAN|avg ACC|
> > > |-|-|-|-|-|-|-|-|-|-|-|
> > > |ACC|90.20|90.20|91.19|89.74|83.00|69.21|89.99|83.20|88.06|86.09|
> > >
> > > These experiments substantiate that our threshold selection remains effective even with single-source validation data. We have incorporated these analyses into appendix to prevent potential misunderstandings.
> > >
> > > Your insightful critique has significantly strengthened our work, and we deeply appreciate your effort throughout this review process!

---

> > > > ### Comment · Reviewer_9kfu · 2025-08-05
> > > > **Strong rebuttal**
> > > >
> > > > The authors addressed all of my concerns and are adding new experimental results accordingly to their paper (or the appendix). Therefore, I think the paper is in a better shape now and I raise my rating.

---

> > > > > ### Author Response · Authors · 2025-08-06
> > > > >
> > > > > Thank you for your positive assessment and for raising the rating of our paper!
> > > > >
> > > > > We confirm that all suggested modifications have been carefully implemented in the final version. Your constructive feedback has been invaluable in improving this work!

---

### Official Review · Reviewer_6ymL · 2025-07-01

**Clarity:** 2
**Significance:** 3
**Originality:** 2
**Rating:** 4
**Confidence:** 5

**Summary:**

This paper proposes a novel framework for detecting AI-generated images by applying a tailored energy-based out-of-distribution (OOD) detection method to features extracted from self-supervised vision models such as DINOv2 and CLIP. The authors provide theoretical motivation for using energy-based formulations and evaluate their method across various datasets. The supplementary material contains additional experimental details and theoretical analysis.

**Questions:**

See above

**Ethical Concerns:**

["NO or VERY MINOR ethics concerns only"]

**Final Justification:**

Raise my rating following the extensive rebuttal with additonal experiments and discussion of current work, including favourable copmarisons.

**Limitations:**

See above

**Paper Formatting Concerns:**

no concerns

**Quality:**

3

**Strengths And Weaknesses:**

Strengths:

1.	Novelty: This is the first work to apply an energy-based OOD method specifically to the task of detecting generated images. The use of energy-based detection methods for generated image detection provides a fresh and promising direction.

2.	Theoretical Rigor: The authors provide clear theoretical grounding for their method, offering insight into why energy-based scores should help distinguish real from generated content.

3.	Comprehensive Supplementary Material: The appendix includes relevant additional experiments, ablations, and theoretical elaboration that enhance reproducibility and depth.

Major Weaknesses:

1.	Missing many recent rlevant works and experimental comparisons:

a.	Rigid - Zhiyuan He et al. Rigid: A training-free and model-agnostic framework for robust ai-generated image detection. arXiv preprint arXiv:2405.20112, 2024.  This paper uses DINOv2 and additive Gaussian noise to separate real and generated images and is conceptually close to the proposed work. Despite being unpublished, its relevance makes it appropriate for inclusion in the related work section. A comparison between the simple noise-based approach in Rigid and the energy-based formulation here would strengthen the claim that energy-based scores add meaningful value.

b.	Manifold induce bias detection – J. Brokman et al, “Manifold induced biases for zero-shot and few-shot detection of generated images”, ICLR 2025.  - This is a recent and peer-reviewed zero-shot detection method that predates the submission deadline by approximately four months and offers publicly available code. It is not cited or compared against, which is a critical omission for a paper in this domain.

c.	ZED [7] – Although ZED is briefly mentioned in the related work section, there is no experimental comparison. Given its relevance as a recent zero-shot detection method, a direct comparison is essential to establish the strength and limitations of the proposed approach.

Overall, the absence of experimental comparisons to key recent methods — including ZED, Rigid, and the manifold based detector — makes it difficult to assess the competitiveness and generalizability of this work within the current state of the field. Notably, all three of these methods have been shown to strongly outperform AEROBLADE, as is also demonstrated in this work.

2.	Incorrect experimental framing: The current experiments do not evaluate real vs. generated image detection directly. Instead, they perform OOD detection between generated images and a single real dataset (e.g., ImageNet or LSUN). These datasets are commonly used in OOD detection tasks and primarily reflect domain shifts rather than generation artifacts, making them unsuitable for evaluating real-versus-generated image detection directly. Proper evaluation should treat all real images as in-distribution and generated images as out-of-distribution, regardless of which real dataset is used for calibration. Additional experiments are needed to confirm that the method generalizes across real domains — specifically, by calibrating the in-distribution (ID) on one real dataset and testing on a different real dataset treated as ID.

Minor Weaknesses:

1.	Clarity on novelty vs. prior work in method section: The method section should more explicitly differentiate between components borrowed from prior OOD methods and those introduced in this work. It would also be valuable to evaluate the proposed approach on standard OOD benchmarks, even if the expected performance is low — this would reinforce the claim that it is purpose-built for detecting generated images rather than general OOD tasks.

2.	Pipeline presentation: While the theoretical formulation is clearly written, the practical method pipeline is difficult to follow. The paper would benefit from a visual figure or algorithm block summarizing the full detection process, from feature extraction to energy computation and thresholding. In particular, the role of the batch size N is unclear — since the detection method should operate independently on each image, the use of a batch suggests possible dependence on other samples. If the method compares the input image to negative examples from the ID dataset, this should be explicitly stated, as it also relates to the concerns raised in Major Weakness 2 regarding evaluation protocol.

---

> ### Author Rebuttal · Authors · 2025-07-31
>
> We sincerely appreciate your constructive comments. In response to your feedback, we have provided detailed responses below. We hope our responses address your concerns and contribute to enhancing our work.
>
> **W.1 Missing many recent relevant works and experimental comparisons.**
>
> A.1 Thank you for bringing these outstanding works on zero-shot generated image detection to our attention. Following your kind suggestion, we have conducted an in-depth discussion and experimental comparison(ACC) of these references in our revised version. The experimental comparison between our method and these approaches is presented in the table below. Note that the code of ZED[3] is not publicly available, we are unable to conduct a direct comparison with our approach. Instead, we have included a high-level comparison with this method in the related work section.
>
> First, compared to these zero-shot works, we are the first to formulate generated image detection as an OOD detection task and theoretically demonstrates the generalizability and rationality of using models pre-trained on real images for generated image detection (as seen in RIGID[1], ZED[3], and our work).
>
> Regarding the comparison with RIGID[1], we share conceptual similarities - both utilize the DINOv2 model. However, whereas RIGID employs an intuition-driven noise-based approach, we derive our approach from the training objectives of self-supervised models. This foundational perspective enabled us to design a more effective differential energy scores, achieving better performance.
>
> “Manifold induce bias detection[2]” explores the biases of the implicit probability manifold, captured by a pre-trained diffusion model, which demonstrates both theoretical and experimental rigor. ZED[3] measures how surprising the image under analysis is compared to a model of real images. This closely aligns with our core concept: while ZED distinguishes real from generated images by analyzing the coding cost of lossless compression encoders trained on real images, our approach leverages visual self-supervised models trained on real images – drawing inspiration from human cognition.
>
> Compared to [2], our method shows marginal improvement in experimental results, which demonstrates enhanced generalization capabilities, notably aligning with the theoretical validation of our method's generalizability in Section 4.4. Even when compared to current state-of-the-art zero-shot detection approaches, our method maintains a discernible advantage in generalization capabilities.
>
> ||Midjourney|SD V1.5|ADM|GLIDE|Wukong|VQDM|BigGAN|avg ACC|
> |-|-|-|-|-|-|-|-|-|
> |RIGID[1]|94.07|87.18|51.44|45.91|87.77|52.17|52.98|67.36|
> |Manifold[2]|55.45|63.00|57.27|88.31|65.39|76.86|77.56|69.12|
> |DEnD(ours)|84.44|70.12|90.66|92.23|72.11|92.25|96.91|85.53|
>
> **W.2 Incorrect experimental framing: The current experiments do not evaluate real vs. generated image detection directly. Instead, they perform OOD detection between generated images and a single real dataset (e.g., ImageNet or LSUN). Proper evaluation should treat all real images as in-distribution and generated images as out-of-distribution, regardless of which real dataset is used for calibration. Additional experiments are needed to confirm that the method generalizes across real domains — specifically, by calibrating the in-distribution (ID) on one real dataset and testing on a different real dataset treated as ID.**
>
> A.2 We follow previous works [5,6] to construct the datasets for evaluation when we conduct experiments on GENImage[4] and AIGCbenchmark[6]. However, we agree with your point that proper evaluation should treat all real images as in-distribution and generated images as out-of-distribution, regardless of which real dataset is used for calibration. Thus, we have conducted supplementary experiments to further verify the effectiveness of our method using the rigorous evaluation setting. We would like to highlight that, beyond ImageNet, all real datasets used in our experiments differ from DINOv2's training data.
>
> We randomly sampled subsets from ImageNet, COCO, and LAION datasets to construct a novel test set of real data. After re-evaluating our method with this enhanced dataset, the results(AUROC) demonstrate that our approach maintains exceptional performance even when confronted with significantly more complex real-world scenarios.
>
> ||BigGAN|Mask-GIT|RQ-Transformer|ADM|ADMG|DiT-XL-2|GilaGAN|LDM|styleGAN|avg AUROC|
> |-|-|-|-|-|-|-|-|-|-|-|
> |DEnD|99.53|98.62|98.58|92.71|82.36|70.43|95.58|83.59|91.07|90.28|
>
> **W.3 Clarity on novelty vs. prior work in method section: The method section should more explicitly differentiate between components borrowed from prior OOD methods and those introduced in this work. It would also be valuable to evaluate the proposed approach on standard OOD benchmarks, even if the expected performance is low — this would reinforce the claim that it is purpose-built for detecting generated images rather than general OOD tasks.**
>
> A.3 Thanks for your kind suggestion. In the method section, we first employ established lemmas from OOD detection theory to demonstrate the feasibility of using models trained on real images for generated image detection. We then discuss the limitations of directly applying existing energy-based OOD detection methods. The referenced methods and equations in this discussion are derived from previous works [7][8]. Following this analysis, we introduce our novel approach and present a theoretical theorem on its generalizability. In the revised version, we will add detailed clarifications to avoid potential confusion.
>
> As for evaluate the proposed approach on standard OOD benchmarks, traditional OOD detection distinguishes In-Distribution (ID) and Out-of-Distribution (OOD) samples based on semantic labels (e.g., cats vs. dogs). In contrast, our formulation avoids semantic categorization. As introduced in Section 3.1, we define image patterns of real images as ID and those of synthetic images as OOD. This fundamentally diverges from conventional OOD detection. Consequently, our method is not applicable to traditional OOD detection benchmarks.
>
> **W.4 Pipeline presentation: While the theoretical formulation is clearly written, the practical method pipeline is difficult to follow. The paper would benefit from a visual figure or algorithm block summarizing the full detection process, from feature extraction to energy computation and thresholding. In particular, the role of the batch size N is unclear — since the detection method should operate independently on each image, the use of a batch suggests possible dependence on other samples. If the method compares the input image to negative examples from the ID dataset, this should be explicitly stated, as it also relates to the concerns raised in Major Weakness 2 regarding evaluation protocol.**
>
> A.4 Thank you for your kind suggestion. We will add algorithm block and a visual figure of the full process in the final version to more clearly illustrate our operational procedures. The detailed explanations are as followed.
>
> First, we shuffle the dataset to ensure randomness. For each sample x, we extract its normalized feature from DINOv2, then compute the inner product between this feature and those of the other N−1 samples in the same batch to derive the energy score. Subsequently, we apply random transformations to generate m(x), repeating the same process to obtain the energy score of the transformed sample. The final differential energy score is calculated by taking the difference between these two energy scores.
>
> Regarding the impact of batch size N, as discussed in Section 4.3, the role of batch size N is analogous to the number of negative samples in self-supervised models (Equation 5). We fully concur with your perspective that "the detection method should operate independently on each image," which aligns with our implementation. Before detection, we perform random shuffling of the dataset, ensuring completely stochastic batch selection that does not rely on relationships between specific images. Although our method utilizes inter-sample information within batches during processing, this design preserves sample independence in detection. To address your concern, we conducted multiple experiments on the same dataset. The results demonstrate that the randomness of batch selection does not impact detection performance, confirming that our method does not rely on relationships between specific images.
>
> The statistical significance measures of five repeated experiments on ImageNet are reported below：
>
> AUROC: Mean = 0.9481, Std = 0.00167, 95% CI = [0.94603, 0.95017]
>
> AP: Mean = 0.9333, Std = 0.00293, 95% CI = [0.92966, 0.93694]
>
> [1] Zhiyuan He et al., Rigid: A training-free and model-agnostic framework for robust ai-generated image detection.
>
> [2] J. Brokman et al., Manifold induced biases for zero-shot and few-shot detection of generated images.
>
> [3] Cozzolino et al., Zero-Shot Detection of AI-Generated Images.
>
> [4] Zhu et al., A Million-Scale Benchmark for Detecting AI-Generated Image.
>
> [5] Chen et al., DRCT: Diffusion Reconstruction Contrastive Training towards Universal Detection of Diffusion Generated Images.
>
> [6] Zhong et al., PatchCraft: Exploring Texture Patch for Efficient AI-generated Image Detection.
>
> [7] Fang et al., Is out-of-distribution detection learnable?
>
> [8] Liu et al., Energy-based out-of-distribution detection.

---

### Official Review · Reviewer_scx8 · 2025-07-02

**Clarity:** 3
**Significance:** 2
**Originality:** 3
**Rating:** 4
**Confidence:** 4

**Summary:**

This study presents a generalizable detector, Differential Energy-based Detection (DEnD), for detecting AI-generated images by modeling them as out-of-distribution (OOD) samples relative to natural images, inspired by human cognitive patterns. The authors address the generalization issue common in training-based detectors, which may (often) fail on unseen generative models. They argue that while traditional OOD detection leverages classifiers trained on semantic labels, generated image detection cannot rely on such classifiers due to the absence of consistent semantic labels in synthetic data. Instead, they propose using energy-based detection, grounded in the observation that in-distribution (ID) data typically exhibits lower energy scores than OOD data. To implement this idea, the method incorporates self-supervised learning using a pre-trained model (DINOv2) to capture differences between natural and generated image distributions. The model is optimized to minimize the differential energy score for ID data, detecting deviations in generated samples. The paper offers a theoretical foundation supporting the generalizability of the proposed approach and validates it through experiments. The evaluation spans datasets such as ImageNet and LSUN-BEDROOM, as well as generated image benchmarks including GenImage and AIGCDetectBenchmark, and a video dataset produced by Sora. The proposed DEnD method, evaluated using Average Precision (AP) and Area Under the Curve (AUC), outperforming both training-based and training-free baselines in different datasets.

**Questions:**

While the work has clear motivation, theoretical validation, a new proposal, and thorough experimental evaluation compared to relevant prior studies, the following comments are suggested to be addressed or at least acknowledged by the authors:

a.	In line 214, the statement that “self-supervised models... push down the differential energy” would benefit from clarification. It is recommended to either provide a supporting citation or an explanation of the basis for this claim.
b.	The work appears highly relevant to Liu et al., “Detecting Generated Images by Real Images.” It would be desirable to explicitly compare the proposed method to this work, both in terms of conceptual differences and if possible, through experimental results.
c.	Although not strictly mandatory, it would be informative to include comparative performance on the UnivFD dataset (Ojha et al., “Towards Universal Fake Image Detectors That Generalize Across Generative Models”), which has become one of the standard benchmarks in the field.
d.	While the authors mention that they will release the code upon acceptance, in the rapidly evolving literature landscape, it is preferable to include code, trained weights, and sufficient documentation at the time of submission or during the rebuttal phase.

**Ethical Concerns:**

["NO or VERY MINOR ethics concerns only"]

**Limitations:**

Yes

**Paper Formatting Concerns:**

No paper formatting issues.

**Quality:**

2

**Strengths And Weaknesses:**

a.	Quality: The submission is technically sound, providing a clear problem statement, theoretical proofs, and experimental evaluations on relevant metrics. The claims are well-supported by both theoretical analysis and experimental results. The methods used are appropriate, and the work is presented as a complete and cohesive study. However, the potential for improvement lies in the detection of images generated by certain models (e.g., SD V1.5 and Wukong), where performance drops were observed. The experiments could be further expanded to include other publicly available datasets, such as the UnivFD dataset (Ojha et al., “Towards Universal Fake Image Detectors That Generalize Across Generative Models”). The authors are transparent in evaluating both the strengths and limitations of their work, and it would be a desirable (though not mandatory) enhancement to explicitly include the observed performance drop on images generated by a few generative models as a stated weakness in the paper.

b.	Clarity:
The submission is clearly written and understandable. However, the authors are recommended to refine the following points:
1. At line 139, remove the full stop after “assumptions”.
2. In Table 4, there is a typo: “AEROBLADA”; this should be corrected to “AEROBLADE”.
3. While the paper emphasizes that the model is inspired by human cognitive processes (as mentioned in line 5 and revisited in line 37), this concept is not mentioned again during the model development or in the conclusion. It is desirable to reiterate this point, at least in the conclusion, to help readers stay connected to the central motivation.
4. At line 161, the use of the word “demonstrated” may be reconsidered. Since the corresponding lemmas are theoretically sound but do not empirically guarantee the claims, it might be more appropriate to state that the generalizability is “hypothetically” or “theoretically” validated.
5. Regarding the generated image datasets used in Tables 1 and 2, although the GitHub source is provided in the supplementary material (Appendix G), it is recommended to include a formal citation within the main text. For instance, Stein et al., “Exposing flaws of generative model evaluation metrics and their unfair treatment of diffusion models”, could be cited if appropriate.
6. Although the results section includes qualitative descriptions indicating better performance of the proposed method, it would be helpful to mention specific performance values in the main text where the model clearly outperforms or underperforms compared to baselines.

c.	Significance:
Yes, the results are impactful for the community. While the GitHub code has not yet been released, if it is made available, it has the potential to be used by researchers. The work addresses the relevant task of detecting AI-generated images and demonstrates moderately improved performance compared to previous methods. While it does not introduce a new dataset, it does offer improved detection performance and contributes meaningfully to the field.

d.	Originality:
The work presents a new method for generated image detection by comparing it to several relevant approaches from the forensic detection literature, based on the idea that only real images are used during training. The integration of self-supervised learning via DINOv2 is well-articulated. However, the paper appears to not include a comparison with a relevant method by Liu et al., “Detecting Generated Images by Real Images,” which may offer a meaningful point of reference.

---

> ### Author Rebuttal · Authors · 2025-07-31
>
> We sincerely appreciate the time and effort you have dedicated to reviewing our paper, as well as your constructive comments. In response to your feedback, we have provided detailed responses below, which have been incorporated into our revised version. We hope our responses address your concerns and contribute to enhancing our work.
>
> **W.1 It would be a desirable (though not mandatory) enhancement to explicitly include the observed performance drop on images generated by a few generative models as a stated weakness in the paper.**
>
> A.1 Thanks for your kind suggestion. We agree with your point that maintaining transparency about the limitations of our work is very important. Thus, we have added the following explanations and discussions in the revised paper.
>
> Inspired by the human cognitive process of distinguishing real images from generated ones, we propose using a model pretrained entirely on real images for the task of generated image detection, leveraging the representational capabilities of the self-supervised vision model DINOv2 pretrained on a large corpus of real images. Although we have theoretically proven the generalizability of this approach and demonstrated strong performance across numerous datasets, in practice, due to limitations in the model’s representational capabilities, our method inevitably experiences some performance degradation when dealing with high-fidelity images. For example, our method achieves an accuracy (ACC) of only 71.88% on Stable Diffusion, significantly below the average. When facing highly realistic video models like Sora, although our method outperforms other baselines, its performance is still lower compared to its results on generative models with lower fidelity, such as GANs.
>
> **W.2 It would be informative to include comparative performance on the UnivFD dataset, which has become one of the standard benchmarks in the field.**
>
> A.2 Thank you for your constructive suggestion! We have added the following experiments and discussions to our revised version.
>
> Following previous standard benchmarking settings in the literature [1,2], we also compare our method with baseline methods over the UnivFD dataset [1]. The experimental results demonstrate that our method exhibits remarkable generalization capability, delivering robust performance across multiple benchmarks.
>
> ||UnivFD[1]|NPR[2]|DEnD(ours)|
> |-|-|-|-|
> |ACC|86.9|95.2|92.1|
> |AP|94.5|97.4|98.5|
>
> **W.3 Clarity issues.**
>
> A.3 We sincerely appreciate your meticulous review, patiently pointing out clarity issues, and providing kind suggestions. Accordingly, we have fixed the mentioned clarity issues in the revised version. Namely, we have
> 1. removed the full stop after "assumptions".
> 2. fixed the typo, i.e., "AEROBLADA" to "AEROBLADE".
> 3. highlighted the motivation in the sections of method and conclusion reiterate this point that our method is inspired by human cognitive processes.
> 4. replaced the word "demonstrate" with a more rigorous one, i.e., "theoretically validated".
> 5. included formal citations of [3] in the main text, especially the mentioned Tables 1 and 2. For instance, we have formally mentioned in the method section that Stein et al. pointed out that self-supervised models exhibit superior sensitivity to global characteristics compared to supervised models operating in label spaces. This inspired us to apply self-supervised models to the task of generated image detection.
> 6. highlighted specific performance improvements. For instance, "Compared to the baseline DRCT, our results on the GenImage dataset show a significant improvement of 11.05%, i.e., from 79.39% to 89.37%".
>
> Thank you again for your time and kind suggestions.
>
> **W.4 While the authors mention that they will release the code upon acceptance, in the rapidly evolving literature landscape, it is preferable to include code, trained weights, and sufficient documentation at the time of submission or during the rebuttal phase**
>
> A.4 We will release the code on GitHub with an anonymous link during the discussion phase.
>
> **Q.1 In line 214, the statement that "self-supervised models... push down the differential energy" would benefit from clarification. It is recommended to either provide a supporting citation or an explanation of the basis for this claim.**
>
> A.1 Thanks for pointing out this potentially confusing issue. We have added the following explanations in the revision.
>
> As derived in Eq. 10, the training objective of self-supervised models can be formulated as minimizing the differential energy scores for ID data (real images). Consequently, real images exhibit significantly lower differential energy scores than generated images.
>
> **Q.2 It would be desirable to explicitly compare the proposed method to a previous one [4], both in terms of conceptual differences and if possible, through experimental results.**
>
> A.2 Thanks for your constructive suggestion! We agree that the difference between ours and the outstanding work [4] should be clarified. Thus, we have added the following discussions (in sections of related work and experiment) and results in our revision.
>
> A recent outstanding work also highlights the critical importance of real images in generated image detection [4]. We highlight that the difference between our work and [4] is twofold: 1) The detection process in [4] is modeled as a one-class classification problem, utilizing noise features and frequency domain analysis for detection, whereas our approach models the generated image detection task as an out-of-distribution (OOD) detection task and designs a differential energy score for detection based on the learning objectives of self-supervised models and 2) The actual training process in [4] requires a small number of real images and no generated images, while our method is zero-shot, requiring only a pre-trained model for detection. Moreover, to highlight the difference, we compared our method with [4] on GENImage[5]. The results show that our method can outperform the one-class classification approach.
>
> ||Detect real[4]|DEnD(ours)|
> |-|-|-|
> |ACC|72.46|89.37|
>
>
> **Q.3 Although not strictly mandatory, it would be informative to include comparative performance on the UnivFD dataset (Ojha et al., “Towards Universal Fake Image Detectors That Generalize Across Generative Models”), which has become one of the standard benchmarks in the field.**
>
> A.3 Please refer to our response to Weakness 2.
>
> **Q.4 it is preferable to include code, trained weights, and sufficient documentation at the time of submission or during the rebuttal phase.**
>
> A.4 We will release the code with an anonymous link.
>
> [1] Ojha et al., Towards Universal Fake Image Detectors That Generalize Across Generative Models.
>
> [2] Tan et al., Rethinking the Up-Sampling Operations in CNN-based Generative Network for
> Generalizable Deepfake Detection.
>
> [3] Stein et al., Exposing flaws of generative model evaluation metrics and their unfair treatment of diffusion models
>
> [4] Liu et al., Detecting Generated Images by Real Images.
>
> [5] Zhu et al., A Million-Scale Benchmark for Detecting AI-Generated Image.

---

### Comment · Area_Chair_gbwu · 2025-08-06
**[General Reminder for Authors and Reviewers] Author-Reviewer Discussion Phase Ending Soon**

Dear Authors and Reviewers,

As you know, the deadline for author-reviewer discussions has been extended to August 8. If you haven’t done so already, please ensure there are sufficient discussions for both the submission and the rebuttal.

Reviewers, please make sure you complete the mandatory acknowledgment **AND** respond to the authors’ rebuttal, as requested in the email from the program chairs.

Authors, if you feel that any results need to be discussed and clarified, please notify the reviewer. Be concise about the issue you want to discuss.

Your AC

---

### Note · Authors · 2025-08-13

We sincerely appreciate the dedicated efforts of the Area Chair and reviewers in evaluating our work. During rebuttal, their valuable suggestions and favorable feedback significantly enhanced our paper's quality. We have diligently incorporated all recommendations.

First, we express our appreciation to the reviewers for their recognition and support of our work in the following aspects:

1. Novelty and Motivation: The paper presents a fresh perspective with clear motivation.

2. Theoretical Rigor: The paper provide clear theoretical grounding and problem statement for the method. Theoretical proof and the claims are well-supported by both theoretical analysis and experimental results.

3. Significance: The paper does offer improved detection performance and contributes meaningfully to the field.

4. Method: The methods used are appropriate and the integration of self-supervised learning via DINOv2 is well-articulated. The paper eliminates the need for task-specific training and reduces training costs and mitigates overfitting risks.

5. Experimental results: The results show strong performance. The experiments cover a wide range of generative models, including unknown architectures.

6. Written expression: The paper is clear and easy to follow.

The reviewers also provided numerous valuable recommendations that proved instrumental in elevating the quality of our work. We have thoughtfully addressed all reviewer inquiries and carefully implemented **including, but not limited to**, the following revisions in response to their guidance:

1. Conducted an in-depth discussion and experimental comparison of many recent relevant work.

2. Investigated the impact of distribution shift of real data on our method's performance.

3. Conducted a detailed discussion and analysis of the threshold selection.

4. Conducted an in-depth discussion on the performance degradation of our detector when confronted with certain specific generative models.

5. Evaluated our method on novel datasets and video generation models.

All modifications have satisfactorily addressed the reviewers' concerns, with their positive acknowledgment of these improvements and three reviewers accordingly raised their scores based on our responses. All improvements suggested by the reviewers have been incorporated into our final version.

Once again, we express deep appreciation for the reviewers' critical role in enhancing the quality of our paper.

---

### Decision · Program_Chairs · 2025-09-17

**Decision:**

Accept (poster)

**Comment:**

The recommendation is based on the reviewers' comments, the area chair's evaluation, and the author-reviewer discussion.

This paper studies AI-generated image detection through the lens of out-of-distribution (OOD) sample detection with a customized energy function as the detection metric. All reviewers find the studied setting novel, and the results provide new insights. The major concern of the initial submission was the lack of comparisons and discussions on the more recent training-free detection methods (e.g., RIGID, ZED, and the Manifold-based approach). The authors’ rebuttal and the additional results have successfully addressed the major concerns of reviewers. In the post-rebuttal phase, all reviewers were satisfied with the authors’ responses and agreed on the decision of acceptance.

Overall, I recommend acceptance of this submission. I also expect the authors to include the new results and suggested changes during the rebuttal phase in the final version.